## Registered report

cognition/evolution/psychology

cooperation, problem-solving, meta-analysis, coordination, cognition

**Author for correspondence:**
Liam Keeble
e-mail: L.Keeble1@newcastle.ac.uk

# The evolution of coordination: a phylogenetic meta-analysis and systematic review

Liam Keeble[1], Joel C. Wallenberg[2] and Elizabeth E. Price[3]

[1]Henry Wellcome Building, Medical School, Newcastle Upon Tyne NE2 4HH, UK
[2]Percy Building, School of English Literature, Language and Linguistics, and [3]School of Psychology, Newcastle University, Newcastle upon Tyne NE1 7RU, UK

(iD) LK, 0000-0002-8181-5054

To solve many cooperative problems, humans must have evolved the ability to solve physical problems in their environment by coordinating their actions. There have been many studies conducted across multiple different species regarding coordinating abilities. These studies aim to provide data which will help illuminate the evolutionary origins of cooperative problem solving and coordination. However, it is impossible to make firm conclusions about the evolutionary origins of coordinating abilities without a thorough comparative analysis of the existing data. Furthermore, there may be certain aspects of the literature that make it very difficult to confidently address evolutionary and meta-analytic questions. This study aimed to rectify this by using meta-analysis, phylogenetic analysis and systematic review to analyse the data already obtained across multiple studies, and to assess the reliability of this data. We found that many studies did not provide the information necessary for meta-analysis, or were not comparable enough to other studies to be included in analyses, meaning meta-analyses were underpowered or could not be conducted due to low samples of both studies and different species. Overall, we found that many studies reported small positive effects across studies, but the standard errors of these effects frequently traversed zero.

## 1. Introduction

Two major issues in the field of comparative cognition are: (i) that methods can vary across studies measuring the same variables with different species [1] and (ii) that this makes it difficult to conduct thorough, quantitative comparisons between species in

many cognitive tasks. The present study will attempt to demonstrate how these problems can be assessed with modern systematic review, and meta-analytic and phylogenetic methods, while also demonstrating potential issues and downfalls of these methods. We will attempt this by pooling results from studies assessing the abilities of different species to coordinate their actions, and assessing these studies in their reliability and rigour.

## 1.1. The string-pull task

The ability to coordinate actions has been essential to the daily activity and survival of human beings throughout their evolutionary history [2–5]. These abilities have also been discovered to be a part of many an animal's behavioural repertoire [6].

The cooperative string-pull task has been used frequently across a variety of different species to test what factors influence an animal's ability to solve complex coordination problems. To be successful in this task, two individuals must pull on two ends of a string simultaneously to retrieve rewards from a platform. If only one of the individuals pulls on the string without another, the string will come loose and the task will be impossible to solve [7,8]. Variations on this task have since been designed to test different species of animal in their abilities to coordinate their behaviour. For example, the simultaneous button pushing task used with dolphins [9].

One important iteration of this task is the delay test, where a partner is held back from the apparatus for a period of time. To be successful, participants must wait for their partner to arrive at the apparatus before pulling. The species that have been found to be successful in this iteration are chimpanzees [7], Asian elephants [10] and kea [11] among others. Unsuccessful species tested using this paradigm include rooks [12], ravens [13], African grey parrots [14] and domestic dogs [15]. In the simultaneous release iteration of the task (where both participants are introduced to the apparatus at the same time), tolerance of a conspecific being in close proximity is found as a good predictor of cooperative success [7,13,16].

However, some researchers are sceptical as to whether tolerance is the best predictor of success in this task [17]. Some have found that relationship quality or affiliation between the participants is also a good predictor of success both within and across species [7,12,18–21], and other social factors probably do play a fundamental role. But, researchers are yet to assess how these social factors interact, and also whether there are any other social or non-social factors also affecting coordination success. Such findings are yet to be synthesized and compared in a standardized fashion. A meta-analytic approach can standardize measures of success both within and across species, and compare these measures at the appropriate level and with the appropriate measures [22].

However, social factors may not play as fundamental a role in predicting success in the cooperative string-pull as many researchers believe. Success could in fact be dictated by physical problem solving cognition or inhibition alone, and subjects could be using environmental cues to solve the problem of the task [23]. If data from less social species is missing from the comparative literature, then it is impossible to assess the extent to which social factors, as opposed to non-social factors, are responsible for success in the ability to coordinate actions. It is therefore important to assess bias towards testing more social species in the studies using this task. This would provide an empirical demonstration that researchers should be testing a wider range of species from a variety of different social environments, and a stronger foundation from which to direct future research.

Recently, a review of this literature has suggested that there is a bias towards testing social animals using this paradigm [24], and the lack of data concerning the performance of less social species makes it difficult to ascertain the effects that living in large groups and having tolerant relationships has on the evolution of abilities to coordinate actions. This may also lead to (or, alternatively, be the result of) a publication bias towards successful species. Similar biases may also be found in the way different species are tested, the amount of trials in which they are trained on the task, and the number of trials in which they are tested on the task [24]. There is little consensus as to the evolutionary consequences of certain factors (social group size, diet and brain size) on the evolution of the ability to solve coordination problems. This is largely due to there being little quantitative comparison between species. Without such data, it is impossible to draw any conclusions regarding the selective pressures that contribute to the evolution of coordinating abilities.

As such, a systematic review and quantitative meta-analysis are necessary to standardize and rigorously estimate the effects of different methodological approaches across species, to review the methodological rigour and differences across studies, and to attempt to demonstrate how analyses may be conducted if reliable data is present.

## 1.2. Challenges of analysis

There are several issues in the comparative cognition literature that make drawing reliable conclusions from comparative and/or meta-analytic studies potentially difficult [1]. Differences in the level of analysis (group success or individual success) can mean studies are difficult to compare directly, and can vary the conclusions researchers draw from their results; different sample sizes can make results more or less reliable; and the methods (e.g. number of training and test trials, and apparatus) used by different researchers can make certain subjects, and thus species, more or less likely to be successful.

There is a lot of heterogeneity in the use of the cooperative string-pull task when testing the coordination abilities of different species [24], meaning results are difficult to compare, and studies are rarely of the same quality. Systematic review and, more specifically, quality assessment are useful tools for assessing the quality of studies, where quality is frequently defined as how well a study's methods are matched to its goals [25]. Although quality assessment tools are most frequently used to assess whether results from studies testing for the same effect can be synthesized, they also provide a method for assessing whether studies are comparable, especially where the overall goals of a field, and thus the goals of individual studies in that field, are to make comparative assessments possible. These methods are therefore useful for assuring the comparability of results in the field of comparative cognition/psychology. Although comparisons between species in a single task will not provide a definitive answer to the evolution of coordination, tasks do provide a useful unit for comparison. Many questions in the field will eventually need to compare the performance of species in the same (or a very similar) task, using the same (or a very similar) design, and using a standardized measure of success in that task to attain answers, and future meta-analyses can synthesize evidence along different dimensions to bolster bodies of evidence further. But, doing so requires that individual studies ensure high levels of comparability in order to achieve high quality. However, the quality of existing literature has yet to be extensively assessed in these terms.

Due to large heterogeneity between study designs and a lack of comparable data, meta- and phylogenetic analysis is, at present, going to be difficult for coordination tasks. Thus, it is essential that the sources of such heterogeneity and the effects this has on conducting meta- and phylogenetic analyses need to be assessed. Quality assessment can identify sources of heterogenity [25], and piloting meta-analytic and phylogenetic analysis on available data can identify the effects such heterogeneity can have on analysis. Conducting such analyses rarely leads to completely firm conclusions. In fact, their primary usefulness is frequently derived from their ability to identify gaps and sources of bias in existing literature [26]. Without identifying these gaps and sources of bias, it remains difficult to identify what studies may need to be conducted in the future to make reliable comparative analyses possible, where existing studies may need to be modified and repeated, and what sources of heterogeneity in design need attention when designing those new and repeated studies.

The most productive method for combating these issues is to begin constructing a dataset, and piloting possible analyses on existing data, while also assessing the quality of that existing data. This will allow gaps in the literature, unreliable studies and plausible analyses to be identified. Thus, the aims of the present study are as follows:

— To assess the quality of studies using animal coordination tasks.
— To assess the comparability of these same studies.
— To generate a comprehensive dataset of results from these studies.
— To identify any form of bias in these studies.
— And to determine the plausibility of the application of meta-analysis and phylogenetic analysis to data from these studies.

The present study will assess the quality of, and pool the results from, studies using coordination tasks. Tests will be conducted to estimate the effects of publication bias and methodological differences (e.g. level of analysis, sample size, length of rope, and number of training and test trials) across studies, and to address some of the following prescient questions regarding differences in coordination success across species using meta- and phylogenetic analysis:

— Does number of trials predict success in a simultaneous-release coordination task?
— Do social dynamics (affiliation and tolerance) predict success in a simultaneous-release coordination task?
— Do brain size, social group size or dietary factors predict success in a delayed-release coordination task?

It is unlikely that analyses for the final question will be possible given available data. However, data will still be collected towards this question in the hope that this will identify where future studies can be conducted to make such analyses plausible for the field. Despite limited available data, some data will allow the precision of such analyses to be assessed given available data.

Such a dataset of studies will be highly beneficial to the field, making it easier for researchers to direct their efforts towards gaps in the literature, and hopefully making it possible to assess evolutionary questions with reliable data in the future [26]. As such, the present study will extract data for some statistical analyses, but will also be used as an opportunity to extract some data that will not be used for analysis, but will help researchers to identify gaps in the literature and design studies that will be comparable to existing results for future comparative analysis.

# 2. Methods

## 2.1. Search protocol

All searches will be conducted using Google Scholar. Papers will be extracted from the first 50 pages returned from each search with the following truncated terms:

— Cooperat* string-pull
— Cooperat* problem solv*
— Cooperat* partner choice
— Tolera* and cooperat*
— Coordinat* task
— Tolerat* and coordinat*

Each of these phrases will also be used again, but with the instances of 'cooperat*' spelt 'co-operat*', and 'coordinat*' spelt 'co-ordinat*', so that the search protocol accounts for the alternative spellings of the terms. The search using the phrase 'Cooperat* string-pull' will be repeated once again but with 'string-pull' written as 'string pull', in case the term is used without the hyphen in any of the literature.

This search protocol will be repeated in the databases of the journals *Animal Behaviour*, *Animal Cognition*, and *Animal Behaviour and Cognition* in case the searches in Google Scholar do not return all relevant results.

All studies which cite Melis *et al.* [7], Hirata & Fuwa [8] or Crawford [27] according to Google Scholar will also be included.

After these searches are complete, and following PRISMA guidelines [28], all duplicates of studies will be removed, and then abstracts will be screened for studies that are not relevant to the questions of the present review. These non-relevant studies will also be removed from the dataset. The remaining studies will be read, and those that do not meet the inclusion criteria will also be removed.

Finally, the best method of assessing publication bias is to compare unpublished results with published results [29]. The principal investigators (traditionally the final author) of all studies found using the above search protocol will be contacted in an attempt to gather unpublished results for comparison.

## 2.2. Criteria for inclusion

All forms of publication that searches return that fit the following criteria will be included in the analysis (including theses and conference abstracts). However, sources that are not published in academic journals will be excluded from most analyses of publication bias.

To be included in the present meta-analysis, studies must use an apparatus that requires two individuals to coordinate their behaviour in order to be successful, and must also assure that individuals cannot receive a reward without coordinating their behaviour (e.g. participate in an individual task for reward). This is a necessary generalization for the comparative element of the study [30], given that a string-pull apparatus is difficult to replicate with certain species that may survive in water and/or lack the anatomical necessities to operate such an apparatus (e.g. dolphins). The apparatus used in a study must also have a mechanism whereby if one individual attempts the task without another then the task is failed, or researchers must have deemed one individual attempting the apparatus without another as failure.

Studies must also include at least one of the following measures: a measure of the relationship between tolerance of individuals in a dyad and their success in a simultaneous-release coordination

task; a measure of success in a delay-release coordination task; a measure of the relationship between relationship quality/affiliation of individuals in a dyad and success in a simultaneous-release coordination task; or a correlation measure of number of trials and success in a simultaneous-release coordination task.

When testing for a relationship between social dynamics and coordination success, tolerance and affiliation between a dyad can be measured either using observational or experimental methods, but cooperative success must be tested using a simultaneous release string-pull task, where both individuals are introduced to the apparatus at the same time. Tolerance is frequently measured by assessing how willing two individuals are to share food/eat in close proximity [7,12,13,31]. Affiliation is frequently assessed using observational data (e.g. how frequently two individuals are near each other, or how frequently two individuals exhibit affiliative behaviours towards each other) [18–21].

Studies providing measures of the relationship between trials and coordination success must have tested subjects in a simultaneous-release string-pull task, and the number of trials used must include both number of test and training trials received by an individual.

## 2.3. Quality assessment

Quality assessment of included studies will take place before data extraction. A quality assessment tool has been created that takes items from the checklist in appendix G of the *Methods for the development of NICE public health guidance* [32]. This checklist forms the basis for assessing the quality of non-randomized studies that report correlations and associations. It has been modified slightly to make it more relevant to the field and studies at hand. We hope that comparative researchers will use and adapt this tool for their own purposes in future systematic reviews. The questions included are:

1. **Population**
   (a) Is the source population well described?
   (b) Do the selected participants or areas represent the eligible population or area? (e.g. were both sexes represented fairly equally? Were a variety of ages represented fairly equally?)
2. **Methods**
   (a) Was the selection of explanatory variables based on a sound theoretical basis?
   (b) How well were likely confounding factors identified and controlled?
   (c) Was the apparatus used appropriate for the species tested?*
3. **Outcomes**
   (a) Were the outcome measures and procedures reliable? (subjective/objective measures, inter-rater reliability)
   (b) Were all the important outcomes assessed?
4. **Analyses**
   (a) Was the study sufficiently powered to detect an intervention effect (if one exists)? (Is power assessed/reported?)
   (b) Were multiple explanatory variables considered in the analyses?
   (c) Were the analytical methods appropriate?
   (d) Was the precision of association given or calculable? Is association meaningful? (Were estimates, standard error, *p*-values, confidence intervals reported?)
5. **Summary**
   (a) Are the study results internally valid (i.e. unbiased)?
   (b) Are the findings generalizable to the source population (i.e. externally valid)?

Answers to the aforementioned questions can be coded on the following scale, taken from appendix G of the *Methods for the development of NICE public health guidance* [32]:

— ++: '… for that particular aspect of study design, the study has been designed or conducted in such a way as to minimize the risk of bias.'
— +: '… either the answer to the checklist question is not clear from the way the study is reported, or that the study may not have addressed all potential sources of bias for that particular aspect of study design.'
— −: '… for those aspects of the study design in which significant sources of bias may persist.'
— Not reported (NR): '… for those aspects in which the study under review fails to report how they have (or might have) been considered.'

— Not applicable (NA): ' … for those study design aspects that are not applicable given the study design under review (for example, allocation concealment would not be applicable for case-control studies).'

Items followed by a '*' are items that are not originally from the *Methods for the development of NICE public health guidance* [32], but have been added by the authors of the present study.

Entries in the checklist will also include study ID, a description of the study design, and the name of the quality assessor.

Studies will not be excluded from the meta-analysis if they are found to have sources of bias during quality assessment. Doing so would probably make meta-analysis impossible due to lack of data. However, leaving these studies in the meta-analysis despite them being judged relatively unreliable will also serve our purpose of illustrating the bias and heterogeneity such studies can introduce into a meta-analysis, and so they will be flagged in the visualizations and discussions of results.

## 2.4. Data extraction

Data extraction will be performed by the lead author. Another author will extract data from 10% of the total studies included in the present study, and inter-extracter agreement will be assessed. Data extraction will also be conducted by the lead author a second time one week after initial extraction. If inter-extracter agreement is low (Cohen's Kappa <0.8) for the first 10%, 30% of all papers will be double coded [33].

The data extracted will include all the relevant measures that allowed a study to be included in the present analysis, alongside the average number of test and training trials per individual, whether or not the study employed repeated measures, the level of analysis (group/individual), the length of the rope used in the task, the sample size, the delay time used in delay-release coordination tasks, the average number of training trials until success, whether the string-pull used is delay-release or simultaneous-release, a description of the apparatus (e.g. string-pull, button-push), whether subjects in a study have participated in a string-pull task in another study, and the conclusion drawn in a delay-release string-pull (were species successful or not?).

If studies provide a measure of success, or a measure of relationship regarding the above variables, but not as a correlation coefficient (where relationships between variables are concerned), or as an odds ratio (where success in the delay task is concerned), then these measures will be converted using functions from the compute.es package [34]; provided, that is, that the relevant information for doing so is made available by authors. If the primary investigators of studies who are contacted provide unpublished data where it is possible to attain effects for the above measures from that data, then the appropriate analysis will be carried out to do so.

## 2.5. Generating the phylogeny

The ape R package will be used to analyse phylogenetic trees [35] constructed using the phangorn R package [36]. Genetic data for mtDNA of species will be downloaded from GenBank [37] using the 'read.GenBank' ape function. Muscle software will be used to align genetic sequences [38] via the phyloch R package [39]. A rooted phylogenetic tree will be constructed ('upgma' phangorn function), and two unrooted phylogenetic trees will be constructed using two different neighbour joining (using the'NJ' and 'BIONJ' functions, respectively) algorithms [40] for comparison.

The phangorn R package will also be used to assess the reliability of trees. Maximum likelihood methods will be used to assess tree reliability, and trees will then be compared using AIC [36,41,42]. The tree with the lowest AIC will be used to generate lambda correlation structures [43,44] to be used in generalized least-squares models for testing evolutionary predictors of success in a coordination task (see 'Statistical analysis' subsection). It is important to note that this is not the best fitting tree with reality, but the tree that fits the data best of those trees compared. Two different trees will be used in final analyses: one which includes all species in the meta-analysis, and one which includes all primates included in the meta-analysis. Since a variety of species of primate are tested using similar paradigms, the data available probably allows for more reliable quantitative comparisons at the within-family level.

Data for brain size, social group size and dietary breadth will be taken from several sources [45–48].

## 2.6. Statistical analysis

Multi-level random-effects meta-analysis models will be conducted using the metafor R package [49]. The 'rma.mv' function will be used to conduct the meta-analysis of results assessing whether the

number of trials correlated with coordination success across studies, whether tolerance correlated with coordination success across studies, whether affiliation correlated with coordination success across studies, and whether there was a similar effect of success in a delay-release coordination task across studies. Following Dougherty & Guillette [50] and Davies *et al.* [22], study, species and phylogeny will be added to these models as random factors.

One potential issue will be with handling data from studies where analysis has been conducted at different levels (individual and group). The above analysis will be repeated but with data from only studies analysing at the individual level, and with data from only the studies analysing at the group level respectively. Where the same individuals have been tested across studies at the individual level, the average of their effect size will be taken and included in the dataset as opposed to including two measures for a single individual.

Study heterogeneity ($\tau$, $Q$) can be used to assess if studies share a common effect size. Usually, random-effects models are able to assess a true effect across studies, even when we assume the effects of different studies may vary. But, when we also expect there to be variation in the true effects of different groups, a subgroup analysis is necessary. In evolutionary meta-analyses, where different species are tested in one task, it might be expected that there will be a different true effect size for each species, as we probably expect some species to be more successful than others in a specific task or paradigm, and this is valuable information for making evolutionary comparisons. Given that different estimates may be expected for studies of different species, true effect size for an individual species will be difficult to estimate by only analysing all studies together. Therefore, species subgroups will be analysed (when the amount of studies per subgroup exceeds two) for effect size wherever possible alongside larger analyses, and species will be tested as a moderator variable in all meta-analyses where evolutionary factors are not analysed as moderator variables. This will hopefully make effect size for different species easier to estimate, and will allow analyses to be conducted regarding the differing levels of success of species. However, it is important to note that the number of studies included in these analyses are likely to be small, and results from analyses should be assessed with this in mind. Despite this, we hope this analysis will serve as a useful demonstration of more powerful analyses that could be conducted in the future.

Publication bias will be analysed in multiple different ways, and assessments for publication bias will be conducted for all studies testing the success of subjects in a delay-release coordination task, and for subgroups of studies testing the success of a single species in a delay-release coordination task if sufficient data is available. *p*-value distributions will be assessed for all studies testing coordination success in a delay-release string-pull task. Differences between unpublished and published results will be assessed using a random effects meta-analysis model with whether the publication was published or not used as a moderator variable, and study, species and phylogeny included as random factors [22,29,50]. Funnel plots and trim and fill methods will be used to assess whether there are any potential studies missing from the dataset (using the 'funnel' and 'trimfill' metafor functions). The 'regtest' function from the metafor package will be used to test for publication biases regarding cooperative understanding with a mixed-effects meta-regression model for plot asymmetry [49]. Orwin's fail-safe $N$ (calculated using the metafor 'fsn' function) will be used to assess the amount of potential studies that would have to exist in the literature, and to have been missed by the present meta-analysis, in order for an effect that does exist to be non-existent in the population [51]. A cumulative meta-analysis (calculated using the metafor 'cumul' function) will be conducted using a fixed-effects model ranked by sample size from largest samples to small. This will assess whether studies with smaller samples—which may mean the effect detected is less accurate—are skewing the results of larger sample studies—which should be more precise [29].

For an evolutionary analysis of good predictors of coordination success, several moderator variables will be tested in separate models using the 'rma.mv' function from the metafor package [49]. These variables will be endocranial volume, population group size and dietary breadth. Each will be used as an individual predictor of success in a delay-release coordination task. Species and study will be included in models as a random factor, given that some species and studies will appear repeatedly in the dataset. Again, this analysis will be repeated but with results from studies analysed at only the individual and group level respectively. This evolutionary analysis will only be conducted if it is deemed feasible after data collection, where feasible means there are more than 10 different species that can be included in analysis. Where conducted, these tests will primarily be used to assess the precision (the size of the confidence interval for estimates) of such models based on available data.

Generalized least-squares models for phylogenetic analysis will be constructed using the 'gls' function from the nlme R package [52]. Brownian motion models of evolution generated from the phylogenetic trees will be fitted to these models to account for phylogenetic relationships between

the species tested. This will be achieved using the 'corPagel' function from the ape R package [35]. The models will use endocranial volume, various measures of population group size and various measures of dietary breadth as individual predictors of success in a delay-release coordination task. Again, if this analysis is not deemed feasible (as defined above) post data collection, then it will not be conducted; and where conducted, it will be primarily used to address the precision (as defined above) of such models with limited data.

$p$-values will be adjusted using the Holm method [53] where tests are: repeated in meta-analytic models and phylogenetic models; conducted to assess differences between multiple species; conducted on all data and then repeated with subsets of results from studies analysing at the individual and the group levels; and where models are repeated to test publication bias.

The 'forest' function from the metafor package will be used to produce forest plots, and the 'funnel' function will be used to produce funnel plots [49].

# 3. Results

The pre-registered protocol was carried out with only one change to the search strategy as described in the following paragraph. The only changes made to analysis code was to stop the code for the analyses that could not be conducted—based on details outlined in the pre-registered methods—from running. The pre-registration can be found at: https://osf.io/v6qb7. The final data and data analysis script can be found at: https://osf.io/hr6ma/.

## 3.1. The dataset

Four hundred and sixty-three studies cited Melis *et al.* [7], 196 studies cited Hirata & Fuwa [8] and 354 studies cited Crawford [27]. The search proposed could not be conducted in the journal *Animal Behaviour* due to the database not accepting the search format (wildcards). Instead, this search was conducted in the journals *Learning and Behaviour*, and *Behavioural Ecology and Sociobiology*. Nine hundred and forty-nine studies were extracted in searches in *Behavioural Ecology and Sociobiology*. Six hundred and sixteenth studies were extracted in searches conducted in the *Animal Cognition* journal database. One hundred and ninety-three studies were extracted from the database of the journal *Learning and Behaviour*.

As shown in figure 1, of the 42 full-text articles assessed for eligibility and removed, 17 were removed because they did not report any of the necessary measures and 17 were removed because participants could still succeed in a task without coordination. Two were removed because the results were repeated in other publications, three because they employed the wrong iteration of the task for the result reported (i.e. they employed a delay task but an estimate of the relationship between affiliation and success, or a simultaneous release iteration and an estimate of coordination success, or used neither a simultaneous or delay release iteration), and three more were removed because they were unaccessible.

Fifty per cent of studies were analysed at the dyadic level, and 50% at the group level. At the group level: 94% were repeated measures; 53% of studies employed both simultaneous string-pull and delay string-pull, and 29% of studies employed just simultaneous, with 18% employing only the delay iteration of the task; 58% of studies claimed their subjects were successful in a task, 23% claimed no success, and the rest were ambiguous or irrelevant; 94% were journal articles, and 94% were published. Of studies that drew a conclusion on the success of their study subjects, 72% claimed their subjects were successful. The dataset included eight studies of birds (seven different species: kea, ravens, rooks, grey parrots, orange-winged parrots, blue-throated macaws and peach-fronted parakeets) and nine studies of mammals (five different species: chimpanzees, dolphins, dogs, marmosets and elephants).

Sixteen authors were contacted via email in an attempt to ascertain the existence of any unpublished datasets that may be relevant to our study. Fourteen authors were initially contacted, and two of those authors recommended we contact another researcher, which we did. If authors did not respond to an initial email, a second email was sent a month after the first. Of the 16 authors contacted, four responded stating that they did not have any unpublished material.

## 3.2. Quality assessment and reliability

Quality assessment finds that while most studies describe the source population well (1a in table 1; numbers and letters in brackets refer to this same table in the present paragraph), studies are frequently not representative of the population at large (1b). The selection of explanatory variables were usually mostly based on recent theory (2a), and confounding factors tended to be well controlled in studies (2b).

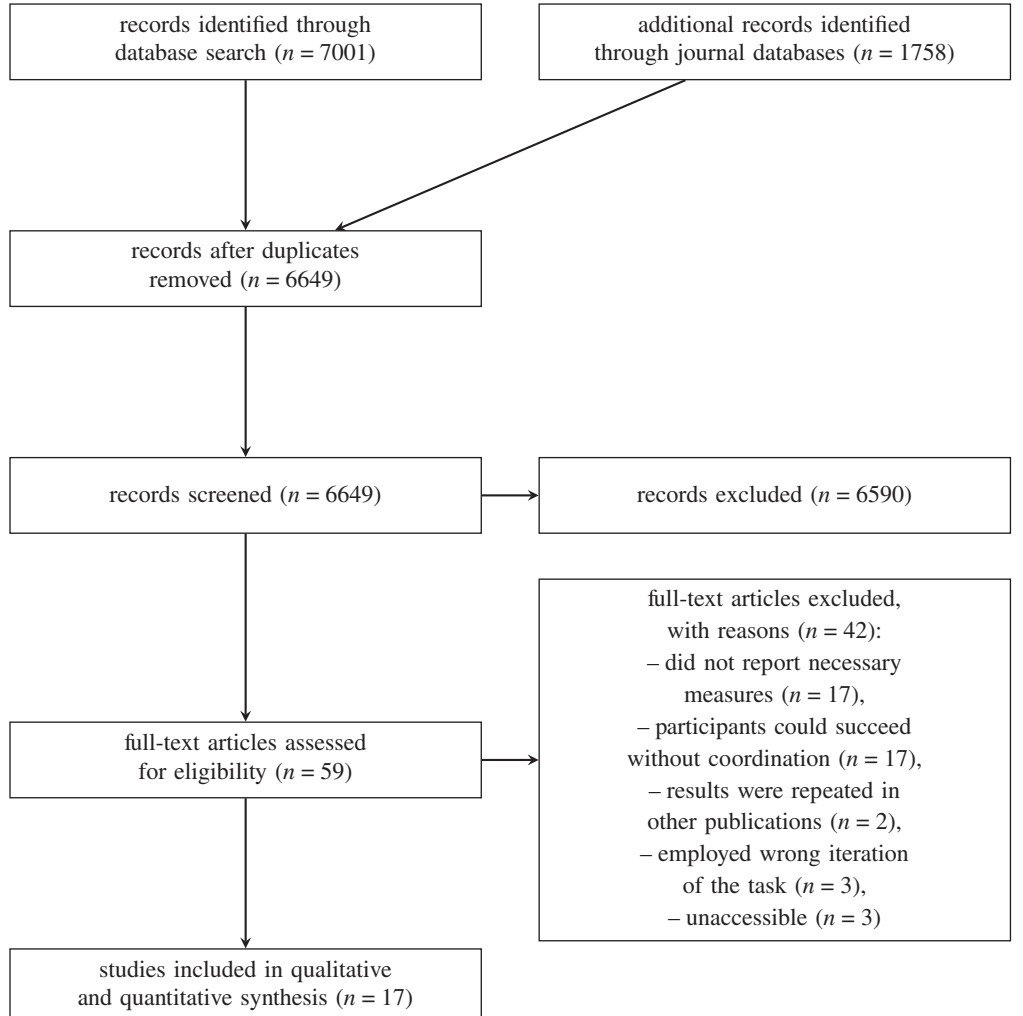

**Figure 1.** A PRISMA diagram of the search protocol.

Apparatus was generally well adapted to the species tested (2c). Studies were generally split in terms of their tests of rater reliability, with half of the studies reporting these tests and half of the studies not reporting them (3a). Studies generally reported all important outcomes, and all important outcomes were generally assessed (3b). Studies rarely reported the power of their study designs (4a), but they generally considered multiple explanatory variables in analyses (4b), and analytical methods were usually appropriate (4c). Studies mostly reported the precision of effect sizes, despite the quality of reports being variable between studies (4d). In summary, quality assessment found that study results were internally valid (5a), but often were not very generalizable to larger populations (5b).

In some cases, where studies used several training conditions, the number of training trials was recorded as the number of trials in which an individual could interact with the string-pull apparatus prior to the trial used to collect data on the result of interest for the meta-analysis. In some cases, some rope lengths and delay times were made incremental in studies, and were recorded in the present dataset as such. Risk ratios were calculated from comparisons of success rates to an imaginary population of the same size with a success rate of 50% to provide effect sizes for success in the delay task that are appropriate for meta-analysis. An external coder (R.W.) coded five papers from the dataset and reliability analyses were conducted. Reliability between coders was mostly high, as illustrated in table 2. However, the sample was not large enough to provide highly reliable estimates of inter-rater similarity.

## 3.3. Phylogenetic trees

Comparisons between phylogenetic trees found that the tree with the lowest AICc was the tree constructed using neighbour joining tree estimation. The difference between the AICc of this tree and the next best (constructed using the unweighted paired group method) was 25.6.

**Table 1.** A table showing how many articles were coded in each category for each quality assessment question.

| question | ++ | + | − | NR |
|---|---|---|---|---|
| *population* | | | | |
| (1a) Is the source population well described? | 6 | 11 | 0 | 0 |
| (1b) Do the selected participants or areas represent the elligible population or area? | 2 | 5 | 10 | 0 |
| *methods* | | | | |
| (2a) Was the selection of explanatory variables based on a sound theoretical basis? | 1 | 15 | 1 | 0 |
| (2b) How well were likely confounding factors identified and controlled? | 1 | 10 | 6 | 0 |
| (2c) Was the apparatus used appropriate for the species tested? | 15 | 2 | 0 | 0 |
| *outcomes* | | | | |
| (3a) Were the outcome measures and procedures reliable? | 8 | 1 | 0 | 8 |
| (3b) Were all the important outcomes assessed? | 12 | 4 | 1 | 0 |
| *analyses* | | | | |
| (4a) Was the study sufficiently powered to detect an intervention effect? | 0 | 0 | 1 | 16 |
| (4b) Were multiple explanatory variables considered in analyses? | 4 | 8 | 5 | 0 |
| (4c) Were the analytical methods appropriate? | 2 | 12 | 3 | 0 |
| (4d) Was the precision of association given or calculable? Is association meaningful? | 3 | 6 | 8 | 0 |
| *summary* | | | | |
| (5a) Are the study results internally valid? | 1 | 13 | 3 | 0 |
| (5b) Are the findings generalizable to the source population? | 0 | 3 | 14 | 0 |

## 3.4. Publication bias

There was substantial heterogeneity between levels of success in studies using the delay iteration of a coordination task ($\tau = 0.2$, $Q = 13.4$, d.f. = 6, adjusted $p = 0.12$). The model found an estimate of success of 0.38 (s.e. = 0.11, $z = 3.47$, adjusted $p = 0.009$), as shown in figure 2. Figure 3 illustrates the change in effect size as the effect sizes of studies using smaller and smaller samples are added to the same meta-analysis. A trim-and-fill method applied to this model finds the same estimate of 0.38 (s.e. = 0.11, $z = 3.47$, adjusted $p = 0.009$). This model also estimated a similar level of heterogeneity when accounting for unpublished results ($\tau = 0.2$, $Q = 13.4$, d.f. = 6, adjusted $p = 0.12$). However, a regression test for funnel plot asymmetry finds a $z$ value of $-3.55$ (adjusted $p = 0.009$), and finds a limit estimate of 1.47 (CI: 0.88–2.06). Similarly, a rank correlation test finds a $\tau$ value of $-0.98$ (adjusted $p = 0.009$), and 0 missing studies. A fail-safe $N$ analysis estimates that no extra studies, with an average effect size of 0.28 and a target effect size of 0.14, would need to be added to the analysis before the effect size shrunk to a trivial value.

No unpublished data were found when authors of studies were contacted via email. Only one unpublished study was identified during searches, meaning there is only one unpublished study in the analysed dataset. A model assessing the difference between published and unpublished studies finds a significant difference between the two (estimate = $-8.86$, s.e. = 2.68, $z = -3.31$, adjusted $p = 0.009$). There is significant heterogeneity between both the moderator ($Q = 10.96$, d.f. = 1, adjusted $p = 0.009$) and studies in general ($Q = 13.4$, d.f. = 6, adjusted $p = 0.12$).

## 3.5. The plausability of meta-analyses

One analysis that could be conducted assessed the relationship between experiment with a task and coordination success. As shown in figure 4, the relationship over all extracted results between

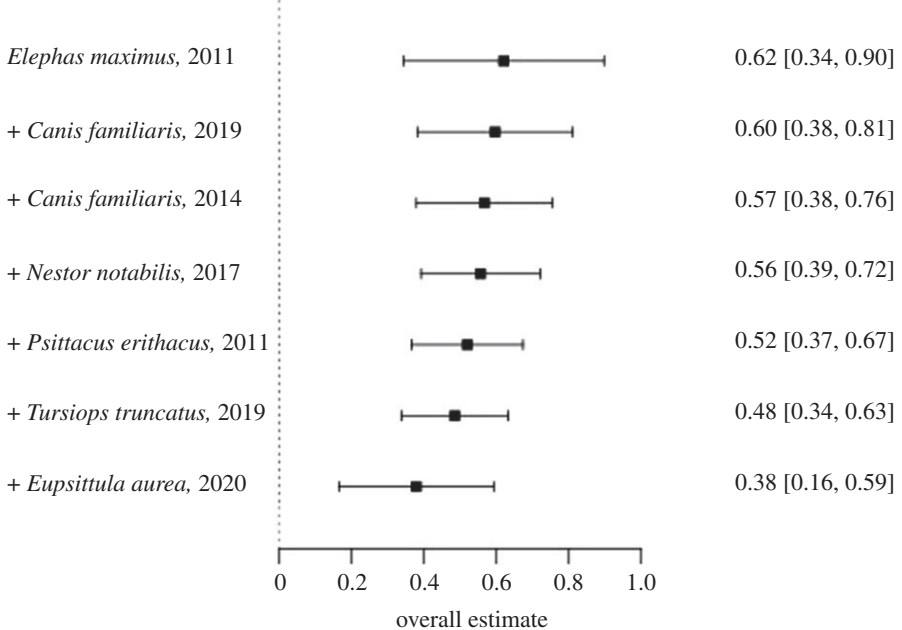

**Figure 2.** Cumulative estimates of the success of different species in a delay iteration of coordination tasks, added by largest to smallest sample sizes.

**Table 2.** Table of results from reliability analysis. Results are proportion of the exact same answers between coders. Note that, for continuous variables, this is often different from the test for reliability, which estimates how closely correlated coders were in their coding of continuous measures as well as agreement between coders. Reliability is the results from the reliability analysis. $\kappa$ values are weighted $\kappa$ values.

| variable | result | reliability |
|---|---|---|
| experience estimate | 4/5 | ICC = 1, $p < 0.001$ |
| $p$-values of delay estimates | 3/5 | ICC = 0, $p = 0.49$ |
| success in delay task estimates | 2/5 | ICC = 1, $p < 0.001$ |
| no. of training trials | 3/5 | ICC = 0.69, $p = 0.047$ |
| no. of test trials | 4/5 | ICC = 1, $p < 0.001$ |
| delay task rope length | 5/5 | $\kappa$ = NA |
| sample size | 4/5 | ICC = 0.71, $p = 0.038$ |
| delay time | 4/5 | ICC = 0.92, $p = 0.0017$ |
| success rate | 3/5 | ICC = 1, $p < 0.001$ |
| trial length | 4/5 | ICC = 0.8, $p = 0.017$ |
| simultaneous and/or delay task | 5/5 | $\kappa$ = 0.64 |
| previous participation | 5/5 | $\kappa$ = 0 |
| conclusion code | 5/5 | $\kappa$ = 1 |
| publication type | 5/5 | $\kappa$ = 1 |
| publication status | 5/5 | $\kappa$ = 0.56 |
| analysis level | 4/5 | $\kappa$ = 0 |
| repeated measures | 5/5 | $\kappa$ = 0 |

experience with the task and success in the task was 0.4 (s.e. = 0.2, $z = 2$, adjusted $p = 0.28$). There was significant heterogeneity ($Q = 40.59$, d.f. = 8, adjusted $p = 0.01$), but a lot of this heterogeneity could be accounted for by including species in the model as a moderator ($Q = 39.95$, d.f. = 5, adjusted $p = 0.01$). When species was included, heterogeneity in effects was reduced ($Q = 0.63$, d.f. = 3, adjusted $p = 1$).

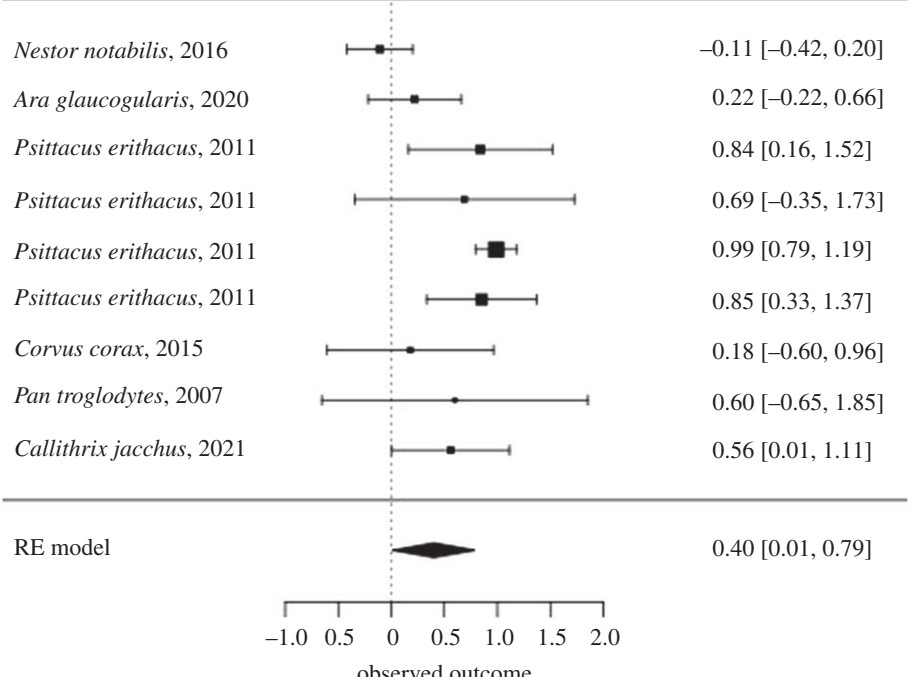

**Figure 3.** A forest plot of estimates for the relationship between experience with a coordination task and success in that task.

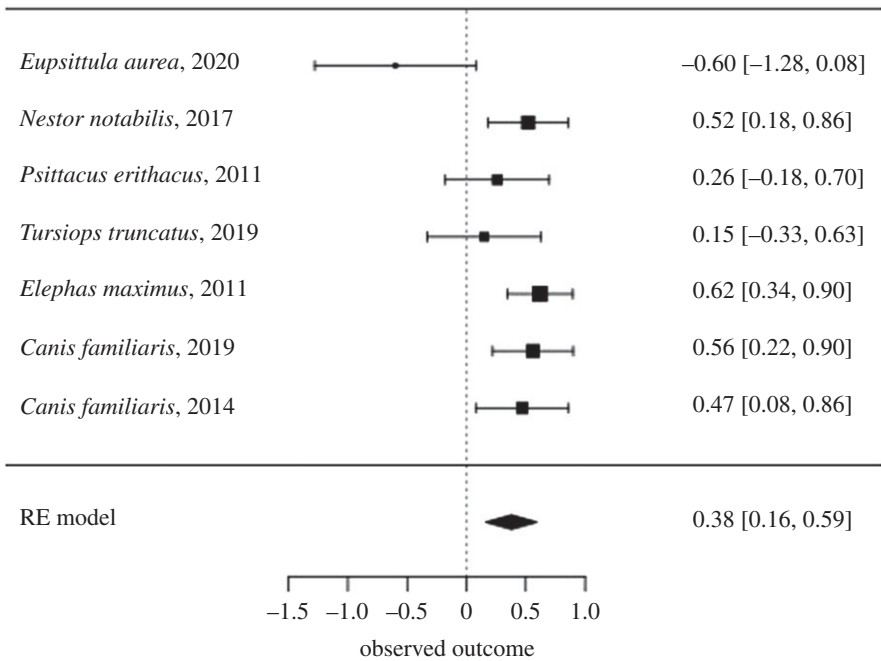

**Figure 4.** Estimates of success for different species in delay iterations of coordination tasks.

When data from different dyads (as opposed to different groups) participating was synthesized, the relationship between experience and success was high (est. = 0.96, s.e. = 0.09, $z = 10.47$, adjusted $p = 0.01$), and heterogeneity in effects was low ($Q = 0.5$, d.f. = 2, adjusted $p = 1$). When analysing results of studies that analysed data at the level of the whole group of subjects, the relationship between experience with the task and cooperative success was 0.29 (s.e. = 0.17, $z = 1.78$, adjusted $p = 0.48$), heterogeneity was estimated at $Q = 9.24$ (d.f. = 5, adjusted $p = 0.5$). Species did not have a significant effect as a moderator in this analysis ($Q = 9.24$, adjusted $p = 0.5$, d.f. = 5). But, when species was included as a moderator variable, heterogeneity between study effect sizes was very low ($Q = 0$, adjusted $p = 1$).

The other models proposed in the pre-registration could not be conducted due to the samples of studies for those analyses being too low. These analyses were assessments of the relationships between cooperative success and tolerance, and cooperative success and affiliation, and meta-analyses of studies analysing at the dyadic level. Furthermore, phylogenetic analyses could not be conducted since the number of different species in analyses never exceeded the required 10 different species. Analyses of individual species could not be conducted due to low frequencies of different studies on any single species.

# 4. Discussion

The present study finds that, while studies of animal coordination are largely internally valid, reports of statistical power and inter-rater reliability are few or variable, and many samples used are not representative of animal populations. Unsurprisingly, where statistical analyses could be conducted, it was found that species differences accounted for a lot of variation in results. Only two of our research questions could be assessed: the relationship between number of trials and success in a coordination task; and the presence of publication bias in the field. The success of most species included in the analysis was correlated with their experience with the coordination task. Statistical tests for publication bias find little evidence of such, despite the one unpublished study that could be included in analyses providing different findings to the published studies. But, these tests may be misleading due to the confirmatory nature of typical assessments of animal cognition.

## 4.1. Comparability between animal coordination studies

Only 17 studies could be included in quality assessment due to many studies not meeting the inclusion criteria. Even then, only some of these studies could be included in certain analyses. The maximum number of studies that could be included in any one analysis was six. This illustrates the disparities in methods, goals and reported outcomes of different studies of animal coordination. Although it may have been possible to widen inclusion criteria, and thus include more studies in some analyses, this would have limited the specificity of the questions that could have been asked. Ideally, future studies should use more standardized study designs when aiming to assess comparative, evolutionary questions, making specific meta-analyses more plausible [46,54]. Tinbergen's four questions can be used to distinguish when studies should be designed similar to others, or can be more flexible in their design [55]. When studies are asking evolutionary questions (as opposed to mechanistic, functional or developmental questions), these can only ultimately be answered by quantitative comparisons. Although qualitative judgements can be useful, quantification of species performance allows a much clearer measure for comparison. Thus comparability in study design and reported results is of the utmost importance when aiming to answer such questions.

## 4.2. Quality assessment and reliability

Quality assessment finds that one author of the present study did not have reasons to doubt the internal validity of most studies of animal coordination assessed. We also find that it is difficult to ensure that studies are externally valid. However, we are limited in the generalizability of our conclusions due to the subjective nature of the assessment. External validity is a known problem in comparative cognition generally [30], and is probably due to the populations of interest being difficult to access. Species are frequently represented in the research literature by small groups of the same, fairly idiosyncratic [50] individuals, and the same is true of the papers assessed in the present study. Furthermore, studies rarely reported statistical power. This may be due to low sample sizes frequently being a practical necessity in the field. However, studies with low sample sizes can still be high powered if appropriate study designs and statistical procedures are used [56]. If samples remain unrepresentative, this may make assessing the larger evolutionary questions exceedingly difficult. Given that the individual rather than the group may be the most reliable unit of interest in assessments of cognition [30,50], comparisons at the species level may be unreliable due to potential individual differences in the population that are not present in the sample. Conducting analyses at both levels will probably produce the most informative and useful results [30] for assessing all levels of evolutionary questions [55].

Inter-rater reliability was high for most coded variables. This is reassuring, given that recent trials of meta research in comparative cognition have been less successful on this front when it comes to coding papers in the field [1]. That the number of training trials had a slightly lower ICC value than most other variables is telling. This is a difficult variable to code for. Training trials varied a lot between both individual subjects within studies and also between studies. Furthermore, since statistical tests of success in a delay task are frequently conducted using various different methods both within and between papers, this may account for the low reliability of $p$-values from those statistical tests. However, given the low sample size in reliability analyses, reliability of these analyses themselves is probably low.

## 4.3. Publication bias

Our statistical tests for publication bias (namely Orwin's fail-safe $N$, a rank correlation test, and the trim-and-fill method applied to the meta-analysis of success measures in the delay task) relied on a small sample of studies which themselves rely on small, heterogeneous samples. Thus we can infer very little about the existence of publication bias in studies assessing animal coordination from our statistical tests. This does not mean that there is no publication bias, but that our sample is simply too small to make any such claims, and that methods of assessing publication bias in a field of such heterogenous studies and samples must be more carefully considered in future. Similarly, the lack of unpublished literature in our dataset does not necessarily mean that such results are non-existent. In fact, it could be a worrying indicator of low retention of datasets and results from unpublished studies, although this claim should be assessed with further research. Overall effect size of success in a delay task was small, and occasionally negative, or standard errors ranged from negative to positive. A cumulative meta-analysis does suggest some pull towards no effect being a result of studies with small samples, and the available unpublished material did report much less successful performances from animals than the published literature (but only one unpublished manuscript was obtained). Unfortunately, a cumulative meta-analysis is limited in this case by the fact that statistical power in the studies assessed is less affected by sample size and more reliant upon repeated measures. As such, the results of the meta-analysis are conservative. Until unpublished literature can be empirically assessed, the presence of publication bias remains uncertain. If anything, the results from statistical tests of publication bias suggest that these tests may be inappropriate for the results tested in the present study. The field is largely based on performing confirmatory tests, with little room for disconfirmatory analyses [57]. For example, statistical analyses of the behaviour of animals in cognitive tasks frequently frame the question in terms of whether there was a significant difference from chance in their behaviour. However, there is no measure of the extent of failure in the actual task, and an animal would be expected to perform at a chance level if they did not understand the task. This means that an animal can only perform at chance or better than chance, but rarely worse. Thus, it would be unsurprising to find that animals quite frequently demonstrate small positive rates of success in the task over many repeated tests despite not really 'understanding' the task. This would be some evidence towards a potential systematic, normally distributed bias towards small, false positive effects. Because many of the effect sizes extracted in the present study were small, statistical tests of publication bias may still find that results are evenly distributed, and thus not skewed (biased) towards a specific result, despite the effects found being small false positives. This is potentially evidence of some systematic bias primarily in study design, rather than at publication.

The above statistical results do clash somewhat with the conclusions drawn by authors. Published authors mostly provided clear conclusions regarding the success or failure of species tested in coordination tasks. This contrasts with recent findings in the field of animal physical cognition, where the conclusions drawn by authors were frequently ambiguous [1]. We also found that the proportion of studies attributing success to species tested in their coordination abilities was similar to that same survey of the conclusions drawn in studies of animal physical cognition [1]. That is, authors in animal physical cognition (when they do draw clear conclusions) are doing so at a rate similar to researchers studying animal coordination. Unfortunately, this uniformity tells us little at present, but may be a fruitful area of research in the future. It could be a signal of systematic bias in study designs, analyses and results, or it could be that the sample of species tested across both domains are simply proportionally good at both coordination and physical tasks, among other potential explanations. The latter is an interesting finding in and of itself. However, if the effect is a signal of bias, this should be checked.

But, despite some authors' claims, the effect sizes extracted from studies of animal coordination did not always correspond to success. Frequently, error values traversed zero, which means evidence may not be extraordinary enough to attribute extraordinary cognitive abilities (such as being able to understand a

complex cooperative problem) to the species tested [30]. This may be further evidence of a systematic bias in study designs due to a lack of methods for disconfirming theory and hypotheses. Note that the rate of successful claims from authors is quite high (72%), despite a high proportion of low positive effect sizes. Although, assessments of other subfields of comparative cognition should assess the presence of such phenomena based on whether the subfield does make use of potentially biased study designs. A change in statistical approach could help to eliminate this bias by assessing the proportion of data in support of success vs. the proportion of data against success, as demonstrated in a recent study with marmosets [58]. There is probably substantial between-species variation in measures of failure at a task. Until such measures of failure are developed and widely applied, highly reliable quantitative comparisons will be difficult. The present study illustrates this difficulty, with a handful of species performing at a similar low level of success.

Phylogenetic (meta) analyses may provide a further potential tonic to such issues for certain questions in comparative cognition and psychology [46,59]. Conducting more comparable studies across species and using data from these studies to perform phylogenetic analyses would allow researchers to assess which species perform better than others in a certain task, as opposed to basing conclusions simply on the success or lack of success of a single species. The complex nature of cognition and factors affecting success in such tasks means that asking whether or not a certain species 'understands' a task is a much more complicated question than a simple binary outcome can answer. However, information regarding which species perform better in a task than others will perhaps provide a more fruitful avenue for understanding factors involved in task success, both evolutionary and cognitive.

## 4.4. The plausability of meta-analyses

One result found in the present study was a relationship between experience with a coordination task and success in that task. This result was fairly consistent across most species included in the analysis, yet there was also heterogeneity across species, as expected. This suggests that all species included in the analysis had mechanisms available to them to decipher successful behavioural strategies for the task, but that some species were more efficient learners than others. However, exactly what was being learned/adapted to cannot be concluded. The high level of success and low level of heterogeneity in the analysis when including only analyses at the dyadic level is probably due to these results coming from a single species (*Psittacus erithacus*). This species showed a high relationship between experience and success. However, in the studies testing this species, subjects were given high amounts of training, and were trained until they had completed the task successfully in 90% of trials. The range of total training trials per subject across studies included in the present study was 1–120, and some could not be specified due to variable numbers of training trials or unspecified total numbers of training trials. This illustrates an obvious bias across species arising from differences in study design. It represents a significant barrier to true quantitative comparisons in the field. Such conclusions can also be affected by researchers analysing their subjects at different levels (analysing per dyad versus analysing over the whole group of subjects). Synthesizing results over all studies (both those analysed at group and dyadic levels) returns a higher overall effect size than analyses conducted at only the group level. Since the meta-analysis treats such analyses at the dyadic level as individual, low-powered studies, if the effects found at the dyadic level are high then they can pull the overall effect size in that direction. This again illustrates the potential effect that individual differences within species can have on potential future phylogenetic meta-analyses [30,50]. This is another potential barrier to making reliable quantitative comparisons between species. When different species are analysed at different levels, it is difficult to assess whether the differences in effect sizes are due to individual or species differences. Thus quantitative inter-species comparison is difficult to interpret in the present analysis. Although analysing at the level of the individual or dyad might be useful for assessing individual differences both within and between species, if authors are attempting to contribute data towards asking evolutionary questions, then analysing at both levels is the most useful. Phylogenetic meta-analyses will require measures of heterogeneity both within and between species to draw any reliable conclusions. At present, large-scale quantitative comparisons of animal coordination remain implausible largely due to variability in study design and level of analysis.

## 4.5. Difficulties of data synthesis

The primary source of difficulty for us when conducting the present study was caused by the reporting of methodological details and results. Coding for the methodological details of studies was often hindered

by researchers detailing these procedures in different ways and across different study designs. For example, it was difficult to determine the number of training and testing trials an animal had participated in, sometimes due to a lack of transparency in methods, but frequently due to some individuals participating in multiple experimental conditions of different kinds. Often, this was despite the fact that such papers were testing the same hypotheses across different species. This was a difficult barrier when trying to determine comparability.

Similarly, large variations in the reporting of statistical results meant that, quite often, effect sizes were not reported or calculable from reported results. No possible way of determining an effect size in these studies meant that many studies that could have been comparable in design could not be compared with those that did report effect sizes and their precision. A lot of this lack of effect sizes can be explained by a large reliance on significance testing statistical practices, which emphasize the reporting of a significance test statistic and *p*-value, with less emphasis on the reporting of effects. Even when we could extract effect sizes, we quite frequently had to calculate many of the effects of importance to us directly from raw data or convert between effect sizes. Standardizing, at least to some extent, the effect size of interest in comparative studies can not only make them more comparable and make meta-analyses more plausible, but can also clarify exactly what researchers are trying to compare across this currently diverse literature. Thus, reporting effect sizes, and reporting them in a clear and reliable fashion, is of the utmost importance for studies in comparative cognition.

## 4.6. Conclusion

At present, results in studies of animal coordination are just not yet comparable enough to conduct reliable phylogenetic analyses. In the small field of animal coordination, studies need to be replicated to ensure validity of conclusions for populations of the samples studied, and more species need to be studied more systematically and with more standardized methods in order to make asking evolutionary questions, quantitative comparisons and phylogenetic analyses possible. This would be the ideal situation, but we recognize that there are extensive practical barriers to such research. Large and varied samples are often difficult to access, especially where species are endangered or difficult to keep in captivity, and studying species in the wild, despite being more ecologically valid, means a loss of experimental control. Comparative cognition as a field at large has faced similar issues in the past [60,61]. Thankfully, some subfields of comparative cognition have been able to make some reliable quantitative comparisions [46,59]. Furthermore, initiatives like the Many Primates project are making such analyses more possible for certain animal taxa [54]. However, to make reliable estimates of the effects of selective pressures versus evolutionary history on certain cognitive traits, data from species from across the evolutionary bush are required [62]. Phylogenetic methods are integral to these efforts. They can be used to synthesize results from both highly controlled experiments and studies with animal groups in the wild. They can also give some added control to studies with wild groups, while also accounting for evolutionary relationships when testing species experimentally.

It is important to continually assess the specific challenges and possibilities that each subfield and comparative paradigm presents. We hope we have demonstrated the importance of this for animal coordination paradigms in the present study. We also hope that we have demonstrated how methods of systematic review (quality assessment in particular), and meta-analysis can be used to assess the extent to which comparative questions can reliably be asked at a given point in time. This will be an ongoing process until it is deemed that there are enough results that are both reliable and similar enough to be synthesized into a phylogenetic analysis. Even then, such quantitative analyses should be continually assessed in this manner to detect (and ultimately mitigate) bias and increase reliability. This is a vital result, as it can be used to direct research in comparative cognition in a reliably productive and specific direction. Thus, at present, comparative and evolutionary questions cannot be asked of animal coordination studies using the data available in the current, sampled literature. Although many of these existing studies can be used to assess other questions of Tinbergen's [55], the evolutionary questions cannot be assessed quantitatively based on the currently available data. Furthermore, large-scale collaborative research can remedy this situation [54,59].

Data accessibility. Data and analysis script for this study is available on the Open Science Framework: https://osf.io/hr6ma/.

Authors' contributions. L.K.: conceptualization, data curation, formal analysis, funding acquisition, investigation, methodology, project administration, visualization, writing—original draft, writing—review and editing; J.C.W.: conceptualization, funding acquisition, methodology, supervision, writing—review and editing; E.E.P.: conceptualization, funding acquisition, methodology, supervision, writing—review and editing.

All authors gave final approval for publication and agreed to be held accountable for the work performed therein.

Competing interests. We declare we have no competing interests.

Funding. This work was supported by the NINE Doctoral Training Partnership.

Acknowledgements. The authors would like to thank Ben Farrar and one other, anonymous reviewer for their very helpful comments on study protocol and the paper itself. We would also like to thank Robin Watson (R.W.) for double coding part of our dataset.

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
