## [Peer Review File · Royal Society Open Science]

Review History

RSOS-201728.R0 (Original submission)

Review form: Reviewer 1

Do you have any ethical concerns with this paper?

No

Recommendation?

Reject

Comments to the Author(s)

Although in principle I applaud the authors' goal of trying to bring together the results of many different cooperation studies with many different species, I am less optimistic this is going to bring forward the field of cooperation in comparative psychology for the reasons I expose below.

In particular, the study proposes the following hypotheses and questions:

(1) Does number of trials predict cooperative success in a delayed partner string-pull paradigm?

The delayed partner string-pull paradigm has produced positive results in a few species (but more species than the ones mentioned in this manuscript). However, this task is not a silver bullet with regards to what we can learn about animals' skills to cooperate. As it has been argued by other researchers (e.g. Seed & Jensen, 2011) there are several simple explanations that can explain how animals learn to pass these tests.

In addition, the number of trials is not the only variable that the different studies have changed. Different studies have also worked with different "delays" and different lengths of the rope, factors which can all influence whether or not and how quickly animals can succeed in this task. Therefore, I am afraid this will be a complicated analysis.

(2) Does inhibition predict cooperative success in a delayed partner string-pull paradigm? For the reasons explained above, there are actually not many species that have succeeded in the delay task under comparable conditions, so the N will be rather low and one would be comparing learning skills under very different conditions. In addition, the inhibition measure would be a species' inhibition measure but not a measure obtained from the same populations that participated in the cooperation tasks. This could be acceptable if everything else was highly controlled. However, it is just another source of noise adding to the other ones, and when dealing with a rather low N.

(3) Do social dynamics (affiliation and tolerance) predict cooperative success? This is possibly the question I am more positive about, just because there are more studies that have looked at tolerance and affiliation. Nevertheless, I would emphasise that cooperation success, as it was measured in many of the studies, does not equate with an understanding of the contingencies and role of the partner in the cooperative endeavour. In other words, one would be looking at something about dyads' relationships and capacity to interact manipulating together food sources but not *collaborative skills* per se.

(4) Do brain size, social group size, or dietary factors predict cooperative understanding? As mentioned above, success in the delay task alone does not mean cooperative understanding. The operationalization of cooperative understanding is more complicated. This study proposal mentions gaze-following too, but that also is not necessarily a good dependent measure: we know that subordinates are often nervous in the presence of dominants when manipulating food rewards, which could lead to more "monitoring" behaviour.

Other comments:

- In the pilot data presented, I noted that Chalmeau 1994 and Suchak et al. 2014 are two of the studies analysed but those studies did not use the same string-pulling task mentioned in the introduction as necessary to facilitate any conclusions about cooperation understanding. The apparatus of those studies were different to the one used in all the delay tasks, so although this may be acceptable, one needs to acknowledge that the contingencies for success with that apparatus are different ones. Individuals can pull alone without losing the opportunity to access the rewards, so waiting in that task is not necessarily needed. If the understanding measure used is "monitoring" behaviour, I am sceptical about it for the reasons mentioned above.

- How informative is the correlation - number of trials and cooperation success? Certainly, with more experience cooperative partners will become more coordinated, but (1) are there significant differences in how quickly different species reach success? (2) Are the levels of success comparable across species? i.e. some species quickly being able to wait for up to 25 sec for the partners whereas others the maximum they wait is 5sec? (3) how is success being measured in

those studies using different pulling apparatuses? Because co-acting is not necessarily cooperating.

Review form: Reviewer 2 (Ben Farrar)

Do you have any ethical concerns with this paper?

No

Recommendation?

Major revision

Comments to the Author(s)

Keeble, Wallenberg and Price propose to meta-analyses and conduct a phylogenetic analysis of data from the co-operative string-pulling literature. I think the proposal is exactly the type of approach the field needs to explore more regarding evidence synthesis, and I think it's great that they've chosen to submit this as a registered report. However, as it stands, I think parts the protocol needs much greater specification, and there are several problems the researchers are likely to face when conducting this meta-analysis which might prevent it from providing strong answers to their questions. Nevertheless, I see a large amount of merit in the proposal. I think that if the authors reframed it more in terms of exploring some of the difficulties in performing comparative meta-analyses, as well as focusing on the substantive questions about co-operation, then it could make a very nice RR. I've framed my comments around each of the proposed analyses:

1. Does number of trials predict cooperative success in a delayed partner string-pull paradigm?

Could the authors specify more about how they will define the number of trials when extracting data, and similarly how they will define success on the string-pulling paradigm. I can see the definition they provide in the caption to Figure 1, but this could be detailed in text.

With trial data, how will the authors deal with differing numbers of training trials and testing trials overall – i.e., if they define the number of trials as the number of testing trials, what happens if studies have markedly different numbers of training trials – or if this information is not available? Similarly, how will the authors address cases where animals are tested sequentially with different partners, potentially across multiple studies, some of which may not have been published? Overall, I think the exact extraction procedure could be specified in greater detail, and the procedure might need to be expanded such that the authors can document the full extent of heterogeneity between the studies. Understanding this heterogeneity is key to understanding what the output of the meta-analyses means – the heterogeneity might be so large that the aggregate point estimates/CIs don't have much meaning – and I imagine most people would have high priors that that more trials leads to greater success anyway. I still think there is a lot of merit in collecting and summarizing this data, and the meta-analysis workflow will help this, but I think the main benefit of this proposal is in collecting and presenting information about individual study results and the between-study heterogeneity.

I have further questions about the models the authors are using for the meta-analysis, and how they will deal with repeated data from individuals (e.g., in the pilot report “Peron et al. 2011 Grey parrots” appears with three separate estimates – if I've found the paper correctly this comes from 3 parrots tested in in 3 consecutive experiments).

As it stands, I don't think the current meta-analysis model accounts for phylogenetic relatedness – they might want to consider fitting a multi-level model which incorporates relatedness in (see e.g. Dougherty & Guillette, 2018 as an example), unless there is a reason why they would prefer to not include this information, but this should be explained.

If the present article goes through a revision, it would be nice if the pilot data, extraction protocol and code could be made available to help understand the procedures and models better.

2. Does inhibition predict cooperative success in a delayed partner string-pull paradigm?

This analysis proposes to see if inhibition scores from MacLean et al.'s study predicts success in the string-pulling. I have strong reservations about the ability of this analysis to have the statistical power to produce a meaningful analysis, most of all because the between-site replicability of the MacLean et al. data is largely untested, and where it has been tested it appears low – i.e., what have been billed as species differences in for example the cylinder task performance may not be so, (see Figure 3 and the following discussion in Farrar et al., 2020). Because the replicability of inhibition data may be low (as well as for the string pulling data) – this analysis might end up predicting noise with noise. I'm not completely opposed to the analysis being performed, but I think it should be heavily caveated if so. If the authors do decide to proceed, I'd encourage them to search further for inhibition data as there are many studies that use the same tasks as MacLean et al., to ensure they are getting as much data as possible to inform the analysis.

I'm also uncertain about what the best way to interpret the results of this analysis are. On the one hand, correlation does not mean causation, and there are likely many reasons why data from the inhibition task will correlate with data from the string pulling task, without there being a causal relationship between them. On the other hand, I think again it's incredibly likely that inhibition is causally related to some aspects of passing the string pulling task. If the authors did not find a positive result then I'd be pretty confident that this would be a false negative (i.e., some relationship between inhibition and string-pulling performance is very likely to exist), and if they do find the positive effect, I'm unsure whether the aggregated numerical estimates would mean very much, because of the heterogeneity between studies.

3. Do social dynamics (affiliation and tolerance) predict cooperative success

4. Do brain size, social group size, or dietary factors predict cooperative understanding?

I have similar reservations about these analyses to the previous two. Again, I think it's highly likely that social dynamics will to some extent predict performance on a social co-operation task (with both causal and non-causal relationships), but it is also possible that the data quality/quantity are not there to perform meaningful quantitative analyses on these. I think the analyses will be useful, but interpreting their outputs may be difficult from a theoretical perspective.

When constructing the phylogenetic trees, how will the authors quantify the uncertainty in the output? They will be able to generate a tree that is the best fit according to some criteria, but if the overall data quality are poor, the tree might be quite likely to be inaccurate, i.e., there is no error control.

My last comments are not specific to a particular analysis:

5. The overall aim of the study

The proposal aims to make “conclusions about the evolutionary origins of cooperative problem solving” by analysing data from the co-operative string-pulling task. However, the generalizability of results from the string-pulling task to other tests (real or hypothetical) of co-operative behaviour is relatively unknown, and as such framing a meta/phylogentic analysis of string-pulling data as an analysis of co-operative behaviour in general is excessive, in my view. The data will be relevant to this question, but not strong. It could instead be framed more narrowly around the string-pulling task.

6. The search protocol and quality control

The search protocol and extraction procedure could be specified to a much a greater extent. One of the strongest outcomes of this study is a high-quality data resource on the string-pulling task. It's important then to ensure that the search and extraction process are comprehensive and high quality. I'd recommend the authors to follow the PRISMA guidelines for reporting, and to add in several quality control stages to their extraction procedure. Regarding the search terms: searching for comparative data can be very difficult due to mass heterogeneity in how researchers report studies, and I think the Google scholar search and citation-based search will identify most studies. However, it would be good to have some verification stages here to check that key studies are not missing. It would be good if the search structure was specified more, too – for example, are the searches simply performed by inputting the four statements into google scholar as they are, or by using truncation and wildcards (e.g. searching for “co-operat*”), will they search for different combinations and spellings (cooperation as well as co-operation). It's possible that the authors have already considered this but it would be great if the full structure of the search could be given. The authors may wish to complement their very general search of the entirety Google scholar with more targeted searches of specialist animal behaviour journals as a method of checking that all relevant publications have been identified.

Could the inclusion criteria be specified further still – will the authors include conference abstracts/thesis chapters/non-published articles? I think they definitely should, but they might want to flag these studies when examining publication bias.

In the extraction process, it is unclear what exactly is being extracted and how. It would be great to build a more comprehensive search protocol, and in particular include details on the study design (training trials, sample sizes, sequential testing etc.) and identify studies in which the participants may overlap. I think it is important to have some form of quality control in this process too, either through double extracting a decent percentage of the studies to assess inter-extractor agreement, or more preferable, to have each extraction checked by another researcher.

7. Detecting publication bias

I think its great the authors are paying attention to publication bias in the analysis, and I think the approach could be expanded to examine publication bias in some of the other data they collect too, e.g., by examining the p-value distributions of reported correlations between tolerance and success on the string pulling task. Currently, their interpretation of the funnel plot and significant Egger's regression test is possibly excessive – a statistically significant asymmetry does not “demonstrate [as in prove] that there are several studies missing”, although it is compatible with that (see e.g. Sterne et al., 2011). There are reasons why comparative datasets from across many species may have asymmetric funnels – for example what if the smaller sample studies - with larger associated errors – are performed on species that are disproportionately likely to pass the test (e.g. a small number of elephants or chimpanzees), whereas the larger sample studies are performed on species that might be less likely to pass the test (e.g. dogs or monkeys). I don't necessarily agree with this reasoning, and think the funnel test is good evidence of publication bias, but currently I believe its overinterpreted. I think it's incredibly difficult to assess publication bias in comparative datasets because of the massive heterogeneity in the laboratories

and species publishing and performing the tasks. I'd encourage the authors to examine subgroups (e.g. by colour coding their funnel plots by species/groups), particularly if they get a decent amount of data on any one particular group.

Finally, the authors may wish to consider alternative methods of detecting publication bias – for example by identifying thesis chapters or conference abstracts that have not been published, or even by simply asking researchers in the field if they have performed any string-pulling tasks that have gone unpublished. Some information like this would complement the statistical analysis massively, in my opinion.

Overall, I think this will be a really valuable study. I think the first aim should be in creating a well-documented data resource around the string pulling task, from which they will be able to assess how much information the meta-analyses and phylogenetic analysis can provide. There are several key problems to navigate for the analyses (heterogeneity, structure of the models, data quality) and it's tricky to review this without having much of the information about the studies yet. This will only be available after extractions have been completed, and so I think it is really important if IPA is to be given to know the extraction process will be very high quality, and that a critical assessment of the strengths of the meta-analyses will be performed after this, or to have another review/discussion about the proposed analyses after extractions have been completed. If it turns out that the data are too low in quality (or in quantity, or homogeneity) to perform strong analyses, then this information is still very valuable to the field, and the process by which the authors will decide this will be a useful methodological advance in the field.

Sincerely,

Ben Farrar

Disclaimer: all my reviews are signed and I am happy to provide clarifications on any points that might be ambiguous to the editors/reviewers/authors (bgf22@cam.ac.uk).

Dougherty, L. R., & Guillette, L. M. (2018). Linking personality and cognition: A meta-analysis. *Philosophical Transactions of the Royal Society B: Biological Sciences*, 373(1756), 20170282. <https://doi.org/10.1098/rstb.2017.0282>

Farrar, B., Voudouris, K., & Clayton, N. (2020). Replications, Comparisons, Sampling and the Problem of Representativeness in Animal Behavior and Cognition Research. *PsyArXiv*. <https://doi.org/10.31234/osf.io/2vt4k>

Sterne, J. A. C., Sutton, A. J., Ioannidis, J. P. A., Terrin, N., Jones, D. R., Lau, J., Carpenter, J., Rücker, G., Harbord, R. M., Schmid, C. H., Tetzlaff, J., Deeks, J. J., Peters, J., Macaskill, P., Schwarzer, G., Duval, S., Altman, D. G., Moher, D., & Higgins, J. P. T. (2011). Recommendations for examining and interpreting funnel plot asymmetry in meta-analyses of randomised controlled trials. *BMJ*, 343. <https://doi.org/10.1136/bmj.d4002>

Decision letter (RSOS-201728.R0)

Dear Mr Keeble,

The Editors assigned to your Stage 1 Registered Report ("The evolution of cooperative problem-solving: A phylogenetic meta-analysis") have now received comments from reviewers. We

would like you to revise your paper in accordance with the referee and editors suggestions which can be found below (not including confidential reports to the Editor). Please note this decision does not guarantee eventual acceptance.

Please submit a copy of your revised paper within six weeks (i.e. by 1st December 2020). If we do not hear from you within this time then it will be assumed that the paper has been withdrawn. In exceptional circumstances, extensions may be possible if agreed with the Editorial Office in advance.

When submitting your revised manuscript, you must respond to the comments made by the referees and upload a file "Response to Referees" in "Section 2 - File Upload". Please use this to document how you have responded to the comments, and the adjustments you have made. In order to expedite the processing of the revised manuscript, please be as specific as possible in your response.

Kind regards,
Professor Chris Chambers
Royal Society Open Science
openscience@royalsociety.org

on behalf of Professor Chris Chambers (Registered Reports Editor, Royal Society Open Science)
openscience@royalsociety.org

Associate Editor Comments to Author (Professor Chris Chambers):
Comments to the Author:

Two specialist reviewers have now assessed the manuscript, with both providing very detailed and constructive assessments of the proposal. Both reviewers are also deeply critical, noting a wide range of areas where insufficient methodological detail is provided (including code and data for the pilot analyses) and unaddressed issues that are central in meta-analysis such as methodological heterogeneity. As you will see, the reviewers (and Rev 1 especially) are also skeptical as to whether the approach, as described, is capable of answering most of the research questions.

I am convinced overall by the reviews that this submission has sufficient merit to invite a Major Revision, but I do want to stress that substantial work will be required to achieve Stage 1 IPA, and a revised submission will be returned to both of the reviewers for re-assessment.

Comments to Author:
Reviewer: 1

Comments to the Author(s)

Although in principle I applaud the authors' goal of trying to bring together the results of many different cooperation studies with many different species, I am less optimistic this is going to bring forward the field of cooperation in comparative psychology for the reasons I expose below.

In particular, the study proposes the following hypotheses and questions:

(1) Does number of trials predict cooperative success in a delayed partner string-pull paradigm?

The delayed partner string-pull paradigm has produced positive results in a few species (but more species than the ones mentioned in this manuscript). However, this task is not a silver bullet with regards to what we can learn about animals' skills to cooperate. As it has been argued by other researchers (e.g. Seed & Jensen, 2011) there are several simple explanations that can explain how animals learn to pass these tests.

In addition, the number of trials is not the only variable that the different studies have changed. Different studies have also worked with different "delays" and different lengths of the rope, factors which can all influence whether or not and how quickly animals can succeed in this task. Therefore, I am afraid this will be a complicated analysis.

(2) Does inhibition predict cooperative success in a delayed partner string-pull paradigm? For the reasons explained above, there are actually not many species that have succeeded in the delay task under comparable conditions, so the N will be rather low and one would be comparing learning skills under very different conditions. In addition, the inhibition measure would be a species' inhibition measure but not a measure obtained from the same populations that participated in the cooperation tasks. This could be acceptable if everything else was highly controlled. However, it is just another source of noise adding to the other ones, and when dealing with a rather low N.

(3) Do social dynamics (affiliation and tolerance) predict cooperative success?

This is possibly the question I am more positive about, just because there are more studies that have looked at tolerance and affiliation. Nevertheless, I would emphasise that cooperation success, as it was measured in many of the studies, does not equate with an understanding of the contingencies and role of the partner in the cooperative endeavour. In other words, one would be looking at something about dyads' relationships and capacity to interact manipulating together food sources but not *collaborative skills* per se.

(4) Do brain size, social group size, or dietary factors predict cooperative understanding?

As mentioned above, success in the delay task alone does not mean cooperative understanding. The operationalization of cooperative understanding is more complicated. This study proposal mentions gaze-following too, but that also is not necessarily a good dependent measure: we know that subordinates are often nervous in the presence of dominants when manipulating food rewards, which could lead to more "monitoring" behaviour.

Other comments:

- In the pilot data presented, I noted that Chalmeau 1994 and Suchak et al. 2014 are two of the studies analysed but those studies did not use the same string-pulling task mentioned in the introduction as necessary to facilitate any conclusions about cooperation understanding. The apparatus of those studies were different to the one used in all the delay tasks, so although this may be acceptable, one needs to acknowledge that the contingencies for success with that apparatus are different ones. Individuals can pull alone without losing the opportunity to access the rewards, so waiting in that task is not necessarily needed. If the understanding measure used is "monitoring" behaviour, I am sceptical about it for the reasons mentioned above.

- How informative is the correlation - number of trials and cooperation success? Certainly, with more experience cooperative partners will become more coordinated, but (1) are there significant differences in how quickly different species reach success? (2) Are the levels of success comparable across species? i.e. some species quickly being able to wait for up to 25 sec for the

partners whereas others the maximum they wait is 5sec? (3) how is success being measured in those studies using different pulling apparatuses? Because co-acting is not necessarily cooperating.

Reviewer: 2

Comments to the Author(s)

Keeble, Wallenberg and Price propose to meta-analyses and conduct a phylogenetic analysis of data from the co-operative string-pulling literature. I think the proposal is exactly the type of approach the field needs to explore more regarding evidence synthesis, and I think it's great that they've chosen to submit this as a registered report. However, as it stands, I think parts the protocol needs much greater specification, and there are several problems the researchers are likely to face when conducting this meta-analysis which might prevent it from providing strong answers to their questions. Nevertheless, I see a large amount of merit in the proposal. I think that if the authors reframed it more in terms of exploring some of the difficulties in performing comparative meta-analyses, as well as focusing on the substantive questions about co-operation, then it could make a very nice RR. I've framed my comments around each of the proposed analyses:

1. Does number of trials predict cooperative success in a delayed partner string-pull paradigm?

Could the authors specify more about how they will define the number of trials when extracting data, and similarly how they will define success on the string-pulling paradigm. I can see the definition they provide in the caption to Figure 1, but this could be detailed in text.

With trial data, how will the authors deal with differing numbers of training trials and testing trials overall – i.e., if they define the number of trials as the number of testing trials, what happens if studies have markedly different numbers of training trials – or if this information is not available? Similarly, how will the authors address cases where animals are tested sequentially with different partners, potentially across multiple studies, some of which may not have been published? Overall, I think the exact extraction procedure could be specified in greater detail, and the procedure might need to be expanded such that the authors can document the full extent of heterogeneity between the studies. Understanding this heterogeneity is key to understanding what the output of the meta-analyses means – the heterogeneity might be so large that the aggregate point estimates/CIs don't have much meaning – and I imagine most people would have high priors that that more trials leads to greater success anyway. I still think there is a lot of merit in collecting and summarizing this data, and the meta-analysis workflow will help this, but I think the main benefit of this proposal is in collecting and presenting information about individual study results and the between-study heterogeneity.

I have further questions about the models the authors are using for the meta-analysis, and how they will deal with repeated data from individuals (e.g., in the pilot report “Peron et al. 2011 Grey parrots” appears with three separate estimates – if I've found the paper correctly this comes from 3 parrots tested in in 3 consecutive experiments).

As it stands, I don't think the current meta-analysis model accounts for phylogenetic relatedness – they might want to consider fitting a multi-level model which incorporates relatedness in (see e.g. Dougherty & Guillette, 2018 as an example), unless there is a reason why they would prefer to not include this information, but this should be explained.

If the present article goes through a revision, it would be nice if the pilot data, extraction protocol and code could be made available to help understand the procedures and models better.

2. Does inhibition predict cooperative success in a delayed partner string-pull paradigm?

This analysis proposes to see if inhibition scores from MacLean et al.'s study predicts success in the string-pulling. I have strong reservations about the ability of this analysis to have the statistical power to produce a meaningful analysis, most of all because the between-site replicability of the MacLean et al. data is largely untested, and where it has been tested it appears low – i.e., what have been billed as species differences in for example the cylinder task performance may not be so, (see Figure 3 and the following discussion in Farrar et al., 2020). Because the replicability of inhibition data may be low (as well as for the string pulling data) – this analysis might end up predicting noise with noise. I'm not completely opposed to the analysis being performed, but I think it should be heavily caveated if so. If the authors do decide to proceed, I'd encourage them to search further for inhibition data as there are many studies that use the same tasks as MacLean et al., to ensure they are getting as much data as possible to inform the analysis.

I'm also uncertain about what the best way to interpret the results of this analysis are. On the one hand, correlation does not mean causation, and there are likely many reasons why data from the inhibition task will correlate with data from the string pulling task, without there being a causal relationship between them. On the other hand, I think again it's incredibly likely that inhibition is causally related to some aspects of passing the string pulling task. If the authors did not find a positive result then I'd be pretty confident that this would be a false negative (i.e., some relationship between inhibition and string-pulling performance is very likely to exist), and if they do find the positive effect, I'm unsure whether the aggregated numerical estimates would mean very much, because of the heterogeneity between studies.

3. Do social dynamics (affiliation and tolerance) predict cooperative success

4. Do brain size, social group size, or dietary factors predict cooperative understanding?

I have similar reservations about these analyses to the previous two. Again, I think it's highly likely that social dynamics will to some extent predict performance on a social co-operation task (with both causal and non-causal relationships), but it is also possible that the data quality/quantity are not there to perform meaningful quantitative analyses on these. I think the analyses will be useful, but interpreting their outputs may be difficult from a theoretical perspective.

When constructing the phylogenetic trees, how will the authors quantify the uncertainty in the output? They will be able to generate a tree that is the best fit according to some criteria, but if the overall data quality are poor, the tree might be quite likely to be inaccurate, i.e., there is no error control.

My last comments are not specific to a particular analysis:

5. The overall aim of the study

The proposal aims to make “conclusions about the evolutionary origins of cooperative problem solving” by analysing data from the co-operative string-pulling task. However, the generalizability of results from the string-pulling task to other tests (real or hypothetical) of co-operative behaviour is relatively unknown, and as such framing a meta/phylogenetic analysis of string-pulling data as an analysis of co-operative behaviour in general is excessive, in my view. The data will be relevant to this question, but not strong. It could instead be framed more narrowly around the string-pulling task.

6. The search protocol and quality control

The search protocol and extraction procedure could be specified to a much a greater extent. One of the strongest outcomes of this study is a high-quality data resource on the string-pulling task. It's important then to ensure that the search and extraction process are comprehensive and high quality. I'd recommend the authors to follow the PRISMA guidelines for reporting, and to add in several quality control stages to their extraction procedure. Regarding the search terms: searching for comparative data can be very difficult due to mass heterogeneity in how researchers report studies, and I think the Google scholar search and citation-based search will identify most studies. However, it would be good to have some verification stages here to check that key studies are not missing. It would be good if the search structure was specified more, too – for example, are the searches simply performed by inputting the four statements into google scholar as they are, or by using truncation and wildcards (e.g. searching for “co-operat*”), will they search for different combinations and spellings (cooperation as well as co-operation). It's possible that the authors have already considered this but it would be great if the full structure of the search could be given. The authors may wish to complement their very general search of the entirety Google scholar with more targeted searches of specialist animal behaviour journals as a method of checking that all relevant publications have been identified.

Could the inclusion criteria be specified further still – will the authors include conference abstracts/thesis chapters/non-published articles? I think they definitely should, but they might want to flag these studies when examining publication bias.

In the extraction process, it is unclear what exactly is being extracted and how. It would be great to build a more comprehensive search protocol, and in particular include details on the study design (training trials, sample sizes, sequential testing etc.) and identify studies in which the participants may overlap. I think it is important to have some form of quality control in this process too, either through double extracting a decent percentage of the studies to assess inter-extractor agreement, or more preferable, to have each extraction checked by another researcher.

7. Detecting publication bias

I think its great the authors are paying attention to publication bias in the analysis, and I think the approach could be expanded to examine publication bias in some of the other data they collect too, e.g., by examining the p-value distributions of reported correlations between tolerance and success on the string pulling task. Currently, their interpretation of the funnel plot and significant Egger's regression test is possibly excessive – a statistically significant asymmetry does not “demonstrate [as in prove] that there are several studies missing”, although it is compatible with that (see e.g. Sterne et al., 2011). There are reasons why comparative datasets from across many species may have asymmetric funnels – for example what if the smaller sample studies - with larger associated errors – are performed on species that are disproportionately likely to pass the test (e.g. a small number of elephants or chimpanzees), whereas the larger sample studies are performed on species that might be less likely to pass the test (e.g. dogs or monkeys). I don't necessarily agree with this reasoning, and think the funnel test is good evidence of publication bias, but currently I believe its overinterpreted. I think it's incredibly difficult to assess publication bias in comparative datasets because of the massive heterogeneity in the laboratories and species publishing and performing the tasks. I'd encourage the authors to examine subgroups (e.g. by colour coding their funnel plots by species/groups), particularly if they get a decent amount of data on any one particular group.

Finally, the authors may wish to consider alternative methods of detecting publication bias – for example by identifying thesis chapters or conference abstracts that have not been published, or even by simply asking researchers in the field if they have performed any string-pulling tasks that have gone unpublished. Some information like this would complement the statistical analysis massively, in my opinion.

Overall, I think this will be a really valuable study. I think the first aim should be in creating a well-documented data resource around the string pulling task, from which they will be able to assess how much information the meta-analyses and phylogenetic analysis can provide. There are several key problems to navigate for the analyses (heterogeneity, structure of the models, data quality) and it's tricky to review this without having much of the information about the studies yet. This will only be available after extractions have been completed, and so I think it is really important if IPA is to be given to know the extraction process will be very high quality, and that a critical assessment of the strengths of the meta-analyses will be performed after this, or to have another review/discussion about the proposed analyses after extractions have been completed. If it turns out that the data are too low in quality (or in quantity, or homogeneity) to perform strong analyses, then this information is still very valuable to the field, and the process by which the authors will decide this will be a useful methodological advance in the field.

Sincerely,
Ben Farrar

Disclaimer: all my reviews are signed and I am happy to provide clarifications on any points that might be ambiguous to the editors/reviewers/authors (bgf22@cam.ac.uk).

Dougherty, L. R., & Guillette, L. M. (2018). Linking personality and cognition: A meta-analysis. *Philosophical Transactions of the Royal Society B: Biological Sciences*, 373(1756), 20170282. <https://doi.org/10.1098/rstb.2017.0282>

Farrar, B., Voudouris, K., & Clayton, N. (2020). Replications, Comparisons, Sampling and the Problem of Representativeness in Animal Behavior and Cognition Research. *PsyArXiv*. <https://doi.org/10.31234/osf.io/2vt4k>

Sterne, J. A. C., Sutton, A. J., Ioannidis, J. P. A., Terrin, N., Jones, D. R., Lau, J., Carpenter, J., Rücker, G., Harbord, R. M., Schmid, C. H., Tetzlaff, J., Deeks, J. J., Peters, J., Macaskill, P., Schwarzer, G., Duval, S., Altman, D. G., Moher, D., & Higgins, J. P. T. (2011). Recommendations for examining and interpreting funnel plot asymmetry in meta-analyses of randomised controlled trials. *BMJ*, 343. <https://doi.org/10.1136/bmj.d4002>

Author's Response to Decision Letter for (RSOS-201728.R0)

See Appendix A.

RSOS-201728.R1 (Revision)

Review form: Reviewer 1

Do you have any ethical concerns with this paper?

No

Recommendation?

Reject

Comments to the Author(s)

Although this revised version addresses some of the issues I raised in the previous round, I still remain unconvinced that a phylogenetic meta-analysis in this field of research can currently bring the field forward. The reasons are, as I commented before, the following:

- There are very few species which have passed the delay task (i.e. chimpanzees, elephants, maybe dogs and dolphins), so with such scarce evidence, I am not sure what is the point of a phylogenetic analysis. In addition, it is now suggested to constrain more which studies enter the analysis to make the comparison more valid (due to the many different testing and experimental protocols), but how many studies would we be left with? if currently only 2-3 species have been considered to pass the task?

- In this revised version Cooperation has been substituted with "coordination", but evidence for coordination can only come from studies showing active efforts to synchronise actions in time and space. Therefore, many studies using other cooperation apparatuses are less indicative of coordination (Suchak et al. Mendres and De Waal, 2003..) because subjects can pull indiscriminately until the apparatus moves (co-occurring when another subject has joined). However, if I understand correctly, these different studies would also enter the analysis. I don't think by changing the word to "coordination", authors can circumvent the difficulty of making conclusions about the mechanisms that bring success about in these co-acting tasks.

- The question about the quality of the relationship between subjects and success in a cooperation task is interesting, but it has already been shown experimentally in several species (e.g. Capuchin monkeys, De Waal & Davis, 2003; Tonquean macaques, Petit et al. 1992; Chimpanzees, Melis et al. 2006; Rooks, Seed et al. 2008). These previous studies have shown the effect of the relationship between partners on cooperative problem solving behaviour *within* a species and *across* species.

One last point is that there is a probably not a *single* method or experimental protocol that can best investigate animals' capacity to solve problems cooperatively with others (and coordinate actions). The type of tasks that can be used with primates are maybe not equally suited for birds or dolphins and the other way around. Therefore, the delay task with the string-pulling paradigm should not be seen as the "silver bullet" to understanding Cooperation/Coordination, but instead as one task that together with others may give us some insight into animals' capacity to coordinate actions with each other.

What we urgently need are multiple paradigms and measures for each species and/or the exact same experiment across different species. Instead, what we currently have is one unique method implemented in myriad ways.

Review form: Reviewer 2 (Ben Farrar)

Do you have any ethical concerns with this paper?

No

Recommendation?

Major revision

Comments to the Author(s)

The revised proposal has addressed many of the previous comments, increased the rigour of the proposal and added in several additional analyses. This will be a complicated project, and there is

still work to be done to ensure it would be effective should IPA be given, but I still think it is a useful project.

1. Structure

The proposal would be easier to follow if the introduction was split in two, with a first section outlining the original research goals (meta-analysing string pulling data), and then a section outlining the challenges of doing so, rather than having the challenges spread throughout the introduction. This may be a personal preference, but I think it would improve the article's usability as a resource for meta-analysis in the area, too.

2. Quality control and assessment

The authors have increased the amount of quality control and assessment stages in the revision.

a) Searches

Although the searches are not fully comprehensive (e.g., animal cognition journals such as *The Journal of Comparative Psychology*, *International Journal of Comparative Psychology*, *Behavioural Processes*, *Learning and Behavior* are not included in the journal search), there is enough redundancy across the three searches to ensure that the vast majority of relevant articles will be identified.

b) Extraction

The extraction process, where the articles are extracted twice by the same individual and then another coding 10% seems fine, although if there are inconsistencies across this 10% more papers will need double coding.

c) Quality Checklist

The authors have introduced a quality checklist – a modified NICE quality appraisal checklist. I applaud the use of such a checklist, and think this is an interesting project in itself, so should be retained. However, using the checklist as a basis for inclusion/exclusion in the meta-analysis might be a problem, as I'm unsure how the authors will be able to judge some of the questions, and it is likely that most studies would end up being excluded if the checklist was followed strictly.

For example, for question 1b:

Do the selected participants or areas represent the eligible population or area? (e.g. were both sexes represented fairly equally? Were a variety of ages represented fairly equally?)

How will the authors know what the eligible populations are without contacting every study author to see which animals were available? Often in captivity the ages of the animals are similar, so many studies will not represent a variety of ages fairly.

For question 2b:

How well were likely confounding factors identified and controlled?

This would require the authors to pretty much peer-review each paper, unless the authors wanted to decide on a few key confounds for the string pulling task and check these in each paper.

Performing the checklist will be an interesting exercise, possibly to quantify just how far these studies are from the strong epidemiological studies that the quality appraisal form was designed to identify. However, I doubt it can be used effectively as a basis of inclusion/exclusion unless it heavily modified.

d) The reliability of the phylogenetic tree

AIC will be used to compare competing trees, but this will only compare the fit of the trees relative to each other, and not absolutely, i.e., the tree with the lowest AIC will not necessarily be a good tree, it will just be the best fitting of all the ones compared on the current dataset. I think this needs to be made clearer in the manuscript, particularly given the noise and limited amount of data that will be informing the trees.

4) Meta-analysis/statistical analyses

The meta-analytic model has been updated to be multi-level, which is good as it can attempt to account for phylogenetic relatedness. However, I don't think the current approach, which includes species and genus only (random = ~1 | Paper/Species/Genus), will capture most of this phylogenetic information – it would only be useful where many species come from the same genus.

A couple of examples of how to include phylogeny with the `rma.mv` function can be found in Cinar et al., 2020 (Model 9) and Davies et al., 2020. Both of these provide good descriptions of how they generate the phylogenetic tree (Davies et al. in particular for when distantly related taxa are used), and provide example code in the supporting information. The approach is to include a single random effect of phylogeny, based on a correlation matrix derived from a phylogenetic tree. Both Cinar et al. and Davies et al. use the same APE package to generate these as the current authors are using, so their code should be adaptable.

The authors have a large number of analyses planned, and they could consider how they will control for multiple testing/whether they think it is an issue.

5) Data and code

Thank-you for sharing the data and code. I managed to reproduce the pilot analysis fully and the data file was readable. When the full dataset is generated, it would be useful to have a data-dictionary to accompany the file explain precisely what each column is and how it was coded – as I expect this dataset might be of interest to many people.

Overall, the new proposal is improved, and I still think it is a feasible and worthwhile project. I share Reviewer 1's concerns about the ability of the project to provide strong answers to the original questions of interest. But even if it fails to answer these questions effectively, then highlighting the barriers around synthesizing evidence like would be a very useful paper. The feasibility of these analyses will depend on the quantity and quality of the overall dataset, which we will only know when it has been extracted. Therefore, it would be useful to conduct a feasibility assessment of the different analyses after the data have been extracted but before conducting them. If IPA is given to the project, I expect a longer Stage 2 review may be necessary than with other registered reports. I still think the project is very interesting and will be of interest those interested in string-pulling tasks but also those interested in evidence synthesis in the field.

Ben

Minor point

1. Code reproducibility

Code, line 355 there is a comma after the first value in

```
> ref = c("HM015213",)
```

This should be removed to read

```
ref = c("HM015213")
```

I was unable to run the code of the phylogenetic analysis – but I assume this just hasn't been completed yet.

References:

Cinar, O., Nakagawa, S., & Viechtbauer, W. (2020). Phylogenetic multilevel meta-analysis: A simulation study on the importance of modeling the phylogeny. *EcoEvoRxiv*.

<https://doi.org/10.32942/osf.io/su4zv>

Davies, A. D., Lewis, Z., & Dougherty, L. R. (2020). A meta-analysis of factors influencing the strength of mate-choice copying in animals. *Behavioral Ecology*, 31(6), 1279–1290.

<https://doi.org/10.1093/beheco/araa064>

Decision letter (RSOS-201728.R1)

Dear Mr Keeble,

The Editors assigned to your Stage 1 Registered Report ("The evolution of coordination: A phylogenetic meta-analysis and systematic review") have now received comments from reviewers. We would like you to revise your paper in accordance with the referee and editors suggestions which can be found below (not including confidential reports to the Editor). Please note this decision does not guarantee eventual acceptance.

Please submit a copy of your revised paper within three weeks (i.e. by the 27-Jan-2021). If we do not hear from you within this time then it will be assumed that the paper has been withdrawn. In exceptional circumstances, extensions may be possible if agreed with the Editorial Office in advance. We do not allow multiple rounds of revision so we urge you to make every effort to fully address all of the comments at this stage. If deemed necessary by the Editors, your manuscript will be sent back to one or more of the original reviewers for assessment. If the original reviewers are not available we may invite new reviewers.

When submitting your revised manuscript, you must respond to the comments made by the referees and upload a file "Response to Referees". Please use this to document how you have

responded to the comments, and the adjustments you have made. In order to expedite the processing of the revised manuscript, please be as specific as possible in your response.

on behalf of Professor Chris Chambers (Registered Reports Editor, Royal Society Open Science)
 openscience@royalsociety.org

Associate Editor Comments to Author (Professor Chris Chambers):

Associate Editor: 1

Comments to the Author:

The revised manuscript was returned to the two original reviewers. Both reviewers offer a range of comments on the updated proposal. Reviewer 2 is broadly positive but also notes several methodological issues requiring further clarification. As in the first round, Reviewer 1 is again the more critical of the reviewers, questioning the overarching rationale for the research and the heterogeneity of the studies included in the meta-analysis. These are major issues that will need to be thoroughly addressed to achieve IPA.

Given the progress made in improving the design, and the enthusiasm of Reviewer 2, I want to give the authors a final opportunity to address these concerns before issuing a final Stage 1 editorial decision.

Comments to Author:

Reviewer: 1

Comments to the Author(s)

Although this revised version addresses some of the issues I raised in the previous round, I still remain unconvinced that a phylogenetic meta-analysis in this field of research can currently bring the field forward. The reasons are, as I commented before, the following:

- There are very few species which have passed the delay task (i.e. chimpanzees, elephants, maybe dogs and dolphins), so with such scarce evidence, I am not sure what is the point of a phylogenetic analysis. In addition, it is now suggested to constrain more which studies enter the analysis to make the comparison more valid (due to the many different testing and experimental protocols), but how many studies would we be left with? if currently only 2-3 species have been considered to pass the task?

- In this revised version Cooperation has been substituted with "coordination", but evidence for coordination can only come from studies showing active efforts to synchronise actions in time and space. Therefore, many studies using other cooperation apparatuses are less indicative of coordination (Suchak et al. Mendres and De Waal, 2003..) because subjects can pull indiscriminately until the apparatus moves (co-occurring when another subject has joined). However, if I understand correctly, these different studies would also enter the analysis. I don't think by changing the word to "coordination", authors can circumvent the difficulty of making conclusions about the mechanisms that bring success about in these co-acting tasks.

- The question about the quality of the relationship between subjects and success in a cooperation task is interesting, but it has already been shown experimentally in several species (e.g. Capuchin monkeys, De Waal & Davis, 2003; Tonquean macaques, Petit et al. 1992; Chimpanzees, Melis et al. 2006; Rooks, Seed et al. 2008). These previous studies have shown the effect of the relationship between partners on cooperative problem solving behaviour *within* a species and *across* species.

One last point is that there is probably not a *single* method or experimental protocol that can best investigate animals' capacity to solve problems cooperatively with others (and coordinate actions). The type of tasks that can be used with primates are maybe not equally suited for birds or dolphins and the other way around. Therefore, the delay task with the string-pulling paradigm should not be seen as the "silver bullet" to understanding Cooperation/Coordination, but instead as one task that together with others may give us some insight into animals' capacity to coordinate actions with each other.

What we urgently need are multiple paradigms and measures for each species and/or the exact same experiment across different species. Instead, what we currently have is one unique method implemented in myriad ways.

Reviewer: 2

Comments to the Author(s)

The revised proposal has addressed many of the previous comments, increased the rigour of the proposal and added in several additional analyses. This will be a complicated project, and there is still work to be done to ensure it would be effective should IPA be given, but I still think it is a useful project.

1. Structure

The proposal would be easier to follow if the introduction was split in two, with a first section outlining the original research goals (meta-analysing string pulling data), and then a section outlining the challenges of doing so, rather than having the challenges spread throughout the introduction. This may be a personal preference, but I think it would improve the article's usability as a resource for meta-analysis in the area, too.

2. Quality control and assessment

The authors have increased the amount of quality control and assessment stages in the revision.

a) Searches

Although the searches are not fully comprehensive (e.g., animal cognition journals such as *The Journal of Comparative Psychology*, *International Journal of Comparative Psychology*, *Behavioural Processes*, *Learning and Behavior* are not included in the journal search), there is enough redundancy across the three searches to ensure that the vast majority of relevant articles will be identified.

b) Extraction

The extraction process, where the articles are extracted twice by the same individual and then another coding 10% seems fine, although if there are inconsistencies across this 10% more papers will need double coding.

c) Quality Checklist

The authors have introduced a quality checklist – a modified NICE quality appraisal checklist. I applaud the use of such a checklist, and think this is an interesting project in itself, so should be retained. However, using the checklist as a basis for inclusion/exclusion in the meta-analysis might be a problem, as I'm unsure how the authors will be able to judge some of the questions, and it is likely that most studies would end up being excluded if the checklist was followed strictly.

For example, for question 1b:

Do the selected participants or areas represent the eligible population or area? (e.g. were both sexes represented fairly equally? Were a variety of ages represented fairly equally?)

How will the authors know what the eligible populations are without contacting every study author to see which animals were available? Often in captivity the ages of the animals are similar, so many studies will not represent a variety of ages fairly.

For question 2b:

How well were likely confounding factors identified and controlled?

This would require the authors to pretty much peer-review each paper, unless the authors wanted to decide on a few key confounds for the string pulling task and check these in each paper.

Performing the checklist will be an interesting exercise, possibly to quantify just how far these studies are from the strong epidemiological studies that the quality appraisal form was designed to identify. However, I doubt it can be used effectively as a basis of inclusion/exclusion unless it heavily modified.

d) The reliability of the phylogenetic tree

AIC will be used to compare competing trees, but this will only compare the fit of the trees relative to each other, and not absolutely, i.e., the tree with the lowest AIC will not necessarily be a good tree, it will just be the best fitting of all the ones compared on the current dataset. I think this needs to be made clearer in the manuscript, particularly given the noise and limited amount of data that will be informing the trees.

4) Meta-analysis/statistical analyses

The meta-analytic model has been updated to be multi-level, which is good as it can attempt to account for phylogenetic relatedness. However, I don't think the current approach, which includes species and genus only (random = ~1 | Paper/Species/Genus), will capture most of this phylogenetic information – it would only be useful where many species come from the same genus.

A couple of examples of how to include phylogeny with the `rma.mv` function can be found in Cinar et al., 2020 (Model 9) and Davies et al., 2020. Both of these provide good descriptions of how they generate the phylogenetic tree (Davies et al. in particular for when distantly related taxa are used), and provide example code in the supporting information. The approach is to include a single random effect of phylogeny, based on a correlation matrix derived from a phylogenetic tree. Both Cinar et al. and Davies et al. use the same APE package to generate these as the current authors are using, so their code should be adaptable.

The authors have a large number of analyses planned, and they could consider how they will control for multiple testing/whether they think it is an issue.

5) Data and code

Thank-you for sharing the data and code. I managed to reproduce the pilot analysis fully and the data file was readable. When the full dataset is generated, it would be useful to have a data-dictionary to accompany the file explain precisely what each column is and how it was coded – as I expect this dataset might be of interest to many people.

Overall, the new proposal is improved, and I still think it is a feasible and worthwhile project. I share Reviewer 1's concerns about the ability of the project to provide strong answers to the original questions of interest. But even if it fails to answer these questions effectively, then highlighting the barriers around synthesizing evidence like would be a very useful paper. The feasibility of these analyses will depend on the quantity and quality of the overall dataset, which we will only know when it has been extracted. Therefore, it would be useful to conduct a feasibility assessment of the different analyses after the data have been extracted but before conducting them. If IPA is given to the project, I expect a longer Stage 2 review may be necessary than with other registered reports. I still think the project is very interesting and will be of interest those interested in string-pulling tasks but also those interested in evidence synthesis in the field.

Ben

Minor point

1. Code reproducibility

Code, line 355 there is a comma after the first value in

```
> ref = c("HM015213",)
```

This should be removed to read

```
ref = c("HM015213")
```

I was unable to run the code of the phylogenetic analysis – but I assume this just hasn't been completed yet.

References:

Cinar, O., Nakagawa, S., & Viechtbauer, W. (2020). Phylogenetic multilevel meta-analysis: A simulation study on the importance of modeling the phylogeny. *EcoEvoRxiv*.
<https://doi.org/10.32942/osf.io/su4zv>

Davies, A. D., Lewis, Z., & Dougherty, L. R. (2020). A meta-analysis of factors influencing the strength of mate-choice copying in animals. *Behavioral Ecology*, 31(6), 1279–1290.
<https://doi.org/10.1093/beheco/araa064>

Author's Response to Decision Letter for (RSOS-201728.R1)

See Appendix B.

RSOS-201728.R2

Review form: Reviewer 2 (Ben Farrar)

Do you have any ethical concerns with this paper?

No

Recommendation?

Accept in principle

Comments to the Author(s)

The revised version has addressed all my concerns with the project. I have re-checked the code and the random effects structure now seems appropriate. I missed the Cinar et al. 2020 reference off my last review - here it is in case it's useful for the authors:

Cinar, O., Nakagawa, S., & Viechtbauer, W. (2020, November 23). Phylogenetic multilevel meta-analysis: A simulation study on the importance of modeling the phylogeny. <https://doi.org/10.32942/osf.io/su4zv>

I think the project is well-planned and feasible and will be a useful study for the reasons I outlined in previous reviews. I also agree with nearly all of Reviewer 1's previous comments about the limitations of the existing string-pull data, and the probable ineffectiveness of various methods of combining them. However, I feel this is exactly the type of project that can effectively highlight these problems - problems that generalise across much comparative research. In addition, the systematic review aspect and curated dataset of this current study should be the next best method for comparing these studies and assessing what the data from the string-pull task mean, even if its marred with uncertainty.

Ben

A minor point:

The introduction starts with: "Three major issues in the field of comparative cognition are: 1) that researchers generalise findings from very small samples to a much larger population... the present study will attempt to demonstrate how these problems can be met with modern systematic review, and meta-analytic..."

This risks overstating the problem of small samples and generalizability somewhat - it is a large problem in some areas of compcog, but others seem to do very well with small samples, e.g. animal learning, animal psychophysics. I also think it oversells how well systematic review and meta-analysis can meet this problems -> perhaps "how these problems can be assessed with systematic review and meta-analysis" might be a better phrasing.

Decision letter (RSOS-201728.R2)

Dear Mr Keeble,

On behalf of the Editors, I am pleased to inform you that your Manuscript RSOS-201728.R2 entitled "The evolution of coordination: A phylogenetic meta-analysis and systematic review" has been accepted in principle for publication in Royal Society Open Science subject to minor revision in accordance with the referee and editor suggestions. Please find their comments at the end of this email.

The reviewers and handling editors have recommended publication, but also suggest some minor revisions to your manuscript. Therefore, I invite you to respond to the comments and revise your manuscript.

Please you submit the revised version of your manuscript within 7 days (i.e. by the 11-Feb-2021). If you do not think you will be able to meet this date please let me know immediately.

When submitting your revised manuscript, you will be able to respond to the comments made by the referees and you should upload a file "Response to Referees". You can use this to document any changes you make to the original manuscript. In order to expedite the processing of the revised manuscript, please be as specific as possible in your response to the referees.

Full author guidelines can be found here <https://royalsocietypublishing.org/rsos/registered-reports>.

on behalf of Professor Chris Chambers (Subject Editor, Royal Society Open Science)
openscience@royalsociety.org

Associate Editor Comments to Author (Professor Chris Chambers):

Associate Editor: 1

Comments to the Author:

One of the previous reviewers (Reviewer 2) was available to assess the Stage 1 submission. Based on this reviewer's assessment, and my own reading of the revised manuscript, I think the proposal is now more solid and we can soon move forward with Stage 1 in-principle acceptance (IPA). However, before doing so, please attend to the final point by Reviewer 2 concerning claims about the potential implications of the work. I agree with the reviewer on this point, and although this minor change in phrasing could be addressed at Stage 2, it makes sense to get the Introduction into as final a state as possible to minimise any need for later revisions.

Once the authors have made this change, IPA will be awarded without further in-depth Stage 1 review.

Reviewer comments to Author:

Reviewer: 2

Comments to the Author(s)

The revised version has addressed all my concerns with the project. I have re-checked the code and the random effects structure now seems appropriate. I missed the Cinar et al. 2020 reference off my last review - here it is in case it's useful for the authors:

Cinar, O., Nakagawa, S., & Viechtbauer, W. (2020, November 23). Phylogenetic multilevel meta-analysis: A simulation study on the importance of modeling the phylogeny. <https://doi.org/10.32942/osf.io/su4zv>

I think the project is well-planned and feasible and will be a useful study for the reasons I outlined in previous reviews. I also agree with nearly all of Reviewer 1's previous comments about the limitations of the existing string-pull data, and the probable ineffectiveness of various methods of combining them. However, I feel this is exactly the type of project that can effectively highlight these problems – problems that generalise across much comparative research. In addition, the systematic review aspect and curated dataset of this current study should be the next best method for comparing these studies and assessing what the data from the string-pull task mean, even if its marred with uncertainty.

Ben

A minor point:

The introduction starts with: “Three major issues in the field of comparative cognition are: 1) that researchers generalise findings from very small samples to a much larger population... the present study will attempt to demonstrate how these problems can be met with modern systematic review, and meta-analytic...”

This risks overstating the problem of small samples and generalizability somewhat – it is a large problem in some areas of compcog, but others seem to do very well with small samples, e.g. animal learning, animal psychophysics. I also think it oversells how well systematic review and meta-analysis can meet this problems -> perhaps “how these problems can be assessed with systematic review and meta-analysis” might be a better phrasing.

Author's Response to Decision Letter for (RSOS-201728.R2)

See Appendix C.

Decision letter (RSOS-201728.R3)

Dear Mr Keeble

On behalf of the Editor, I am pleased to inform you that your Manuscript RSOS-201728.R3 entitled "The evolution of coordination: A phylogenetic meta-analysis and systematic review" has been accepted in principle for publication in Royal Society Open Science.

You may now progress to Stage 2 and complete the study as approved. Before commencing data collection we ask that you:

- 1) Update the journal office as to the anticipated completion date of your study.
- 2) Register your approved protocol on the Open Science Framework (<https://osf.io/rr>) or other recognised repository, either publicly or privately under embargo until submission of the Stage 2 manuscript. Please note that a time-stamped, independent registration of the protocol is mandatory under journal policy, and manuscripts that do not conform to this requirement cannot be considered at Stage 2. The protocol should be registered unchanged from its current approved state, with the time-stamp preceding implementation of the approved study design. We strongly recommend using the dedicated Stage 1 RR registration portal at <https://osf.io/rr>

Following completion of your study, we invite you to resubmit your paper for peer review as a Stage 2 Registered Report. Please note that your manuscript can still be rejected for publication at Stage 2 if the Editors consider any of the following conditions to be met:

- The results were unable to test the authors' proposed hypotheses by failing to meet the approved outcome-neutral criteria.
- The authors altered the Introduction, rationale, or hypotheses, as approved in the Stage 1 submission.
- The authors failed to adhere closely to the registered study procedures. Please note that any deviations from the approved procedures must be communicated to the editor immediately for approval, and prior to the completion of data collection. Failure to do so can result in revocation of in-principle acceptance and rejection at Stage 2 (see complete guidelines for further information).
- Any post-hoc (unregistered) analyses were either unjustified, insufficiently caveated, or overly dominant in shaping the authors' conclusions.
- The authors' conclusions were not justified given the data obtained.

We encourage you to read the complete guidelines for authors concerning Stage 2 submissions at <https://royalsocietypublishing.org/rsos/registered-reports#ReviewerGuideRegRep>. Please especially note the requirements for data sharing, reporting the URL of the independently registered protocol, and that withdrawing your manuscript will result in publication of a Withdrawn Registration.

Once again, thank you for submitting your manuscript to Royal Society Open Science and we look forward to receiving your Stage 2 submission. If you have any questions at all, please do not hesitate to get in touch. We look forward to hearing from you shortly with the anticipated submission date for your stage two manuscript.

on behalf of Professor Chris Chambers (Registered Reports Editor, Royal Society Open Science)
openscience@royalsociety.org

Author's Response to Decision Letter for (RSOS-201728.R3)

See Appendices D-F.

RSOS-201728.R4

Review form: Reviewer 2 (Ben Farrar)

Is the manuscript scientifically sound in its present form?

No

Are the interpretations and conclusions justified by the results?

No

Is the language acceptable?

Yes

Do you have any ethical concerns with this paper?

No

Have you any concerns about statistical analyses in this paper?

No

Recommendation?

Major revision

Comments to the Author(s)

The authors have completed the study in-line with the Stage 1 submission, with their noted change to the search after finding Animal Behaviour did not accept wild cards in the search string. This could have been overcome by using a search engine (e.g., Scopus, Web of Science) to perform the searches but the authors might not have had access.

Overall I believe the authors have performed a useful study and generated a useful dataset. However I believe the inferences the authors make, and general structure of the results and discussion could be much improved to ensure the value of the study is communicated effectively and that the interpretation is justified.

Overall Structure

I thought the overall results and discussion section could be made easier to follow, especially because it jumps between methodological challenges, quality control and analyses. Could subheadings be used corresponding to each of the objectives set out in the introduction to make the structure of the results clearer to the reader?

Quality control

The authors provided a well presented dataset of both the raw data and the quality control, but I was surprised not to see the reasons for the decisions in the quality control document not also

included. While the Stage 1 submission didn't explicitly state that the reasons and relevant text for the decisions to the quality control questions would be recorded, I thought this would have been performed. For example, the authors state that populations were rarely representative, but understanding how this decision was made is important for the quality of the dataset (i.e., have a column in the datafile with the same description from the full text extract and a comment on why the decision has been made). If these data weren't extracted, I think it would be really valuable for the authors to extract them and add them to the dataset for each question. Commenting on why the decision has been made for internal validity seems particularly important (and what criteria were screened for this), especially as the authors make the claim that "studies of animal coordination are largely internally valid". This is not necessarily true, rather what the authors did find was that one coder assessing a small number of string pulling tasks did not find reasons to doubt their internal validity.

Conclusions

Another conclusion the authors made surprised me. About publication bias, the authors said their analysis suggested little publication bias in studies assessing animal co-ordination. This surprised me as I know of unpublished studies in this area, and that the statistical power of their analyses to detect publication bias is very low. If their overall sample size is 17 included studies, what number of unpublished similar studies would mean that there is little publication bias? Even 4 or 5 unpublished studies would be a lot here.. If they wanted to make the conclusion about little publication bias then I think power analyses to see the degree of e.g. asymmetry their analyses could detect would be necessary. Alternatively I see no issue in highlighting how difficult it can be to assess publication in very heterogenous datasets with small sample sizes (this is the conclusion I would draw, with a strong prior that some studies will be missing).

Discussion

The authors discussion focuses on co-ordination tasks and what their data can tell us about string-pulling. As the other reviewer at Stage 1 predicted, this might not be so much. I think a section of the discussion should be devoted to the challenges the authors faced in collecting and synthesising data in this project, as I feel this is one of the major contributions it makes.

Minor Comments

Reasons for exclusion that are given in the text (Lines 442 - 450) also be presented in the PRISMA diagram

Could the raw agreement also be included in Table 2, i.e., 5/5 statements extract the same, or 3/5? I think this would be easier for people to understand given the small number of double coded variables.

Could the quality control statements also be written in Table 1

Decision letter (RSOS-201728.R4)

Dear Mr Keeble:

On behalf of the Editor, I am pleased to inform you that your Stage 2 Registered Report RSOS-201728.R4 entitled "The evolution of coordination: A phylogenetic meta-analysis and systematic review" has been deemed suitable for publication in Royal Society Open Science subject to minor revision in accordance with the referee suggestions. Please find the referees' comments at the end of this email.

The reviewers and Subject Editor have recommended publication, but also suggest some minor revisions to your manuscript. We invite you to respond to the comments and revise your manuscript. Below the referees' and Editors' comments (where applicable) we provide additional requirements. Final acceptance of your manuscript is dependent on these requirements being met. We provide guidance below to help you prepare your revision.

Please submit your revised manuscript and required files (see below) no later than 21 days from today's (ie 21-Dec-2021) date. Note: the ScholarOne system will 'lock' if submission of the revision is attempted after the deadline. If you do not think you will be able to meet this deadline please contact the editorial office immediately.

on behalf of Professor Chris Chambers
(Registered Reports Editor, Royal Society Open Science)
openscience@royalsociety.org

Associate Editor Comments to Author (Professor Chris Chambers):

Associate Editor: 1

Comments to the Author:

One of the two original Stage 1 reviewers kindly returned to evaluate the Stage 2 submission, and I have decided that this reviewer's evaluation together with my own reading is sufficient for us to proceed with an interim editorial decision. As you will see, the reviewer is broadly positive about the completed manuscript, while also offering a range of helpful suggestions for revision, including structural changes to the reporting of results, documentation of the data (particularly in relation to the quality assessment), and more in depth consideration of interpretative limitations in the Discussion (as well as methodological lessons learned).

I concur with this assessment and overall feel that you have completed a very careful and thorough investigation. Provided you are able to respond comprehensively to the points raised by the reviewer, final acceptance should be forthcoming without requiring further in-depth review.

Comments to Author:

Reviewer: 2

Comments to the Author(s)

The authors have completed the study in-line with the Stage 1 submission, with their noted change to the search after finding Animal Behaviour did not accept wild cards in the search string. This could have been overcome by using a search engine (e.g., Scopus, Web of Science) to perform the searches but the authors might not have had access.

Overall I believe the authors have performed a useful study and generated a useful dataset. However I believe the inferences the authors make, and general structure of the results and discussion could be much improved to ensure the value of the study is communicated effectively and that the interpretation is justified.

Overall Structure

I thought the overall results and discussion section could be made easier to follow, especially because it jumps between methodological challenges, quality control and analyses. Could subheadings be used corresponding to each of the objectives set out in the introduction to make the structure of the results clearer to the reader?

Quality control

The authors provided a well presented dataset of both the raw data and the quality control, but I was surprised not to see the reasons for the decisions in the quality control document not also included. While the Stage 1 submission didn't explicitly state that the reasons and relevant text for the decisions to the quality control questions would be recorded, I thought this would have been performed. For example, the authors state that populations were rarely representative, but understanding how this decision was made is important for the quality of the dataset (i.e., have a column in the datafile with the same description from the full text extract and a comment on why the decision has been made). If these data weren't extracted, I think it would be really valuable for the authors to extract them and add them to the dataset for each question. Commenting on why the decision has been made for internal validity seems particularly important (and what criteria were screened for this), especially as the authors make the claim that "studies of animal coordination are largely internally valid". This is not necessarily true, rather what the authors did find was that one coder assessing a small number of string pulling tasks did not find reasons to doubt their internal validity.

Conclusions

Another conclusion the authors made surprised me. About publication bias, the authors said their analysis suggested little publication bias in studies assessing animal co-ordination. This surprised me as I know of unpublished studies in this area, and that the statistical power of their analyses to detect publication bias is very low. If their overall sample size is 17 included studies, what number of unpublished similar studies would mean that there is little publication bias? Even 4 or 5 unpublished studies would be a lot here.. If they wanted to make the conclusion about little publication bias then I think power analyses to see the degree of e.g. asymmetry their analyses could detect would be necessary. Alternatively I see no issue in highlighting how difficult it can be to assess publication in very heterogeneous datasets with small sample sizes (this is the conclusion I would draw, with a strong prior that some studies will be missing).

Discussion

The authors discussion focuses on co-ordination tasks and what their data can tell us about string-pulling. As the other reviewer at Stage 1 predicted, this might not be so much. I think a section of the discussion should be devoted to the challenges the authors faced in collecting and synthesising data in this project, as I feel this is one of the major contributions it makes.

Minor Comments

Reasons for exclusion that are given in the text (Lines 442 - 450) also be presented in the PRISMA diagram

Could the raw agreement also be included in Table 2, i.e., 5/5 statements extract the same, or 3/5? I think this would be easier for people to understand given the small number of double coded variables.

Could the quality control statements also be written in Table 1

===PREPARING YOUR MANUSCRIPT===

one version should clearly identify all the changes that have been made (for instance, in coloured highlight, in bold text, or tracked changes);

===PREPARING YOUR REVISION IN SCHOLARONE===

To revise your manuscript, log into <https://mc.manuscriptcentral.com/rsos> and enter your Author Centre - this may be accessed by clicking on "Author" in the dark toolbar at the top of the

page (just below the journal name). You will find your manuscript listed under "Manuscripts with Decisions". Under "Actions", click on "Create a Revision".

-- If you are requesting an article processing charge waiver, you must select the relevant waiver option (if requesting a discretionary waiver, the form should have been uploaded, see 'File upload' above).

-- If you have uploaded any electronic supplementary (ESM) files, please ensure you follow the guidance at <https://royalsociety.org/journals/authors/author-guidelines/#supplementary-material> to include a suitable title and informative caption. An example of appropriate titling and captioning may be found at https://figshare.com/articles/Table_S2_from_Is_there_a_trade-off_between_peak_performance_and_performance_breadth_across_temperatures_for_aerobic_scope_in_teleost_fishes_/3843624.

At the 'Review & submit' step, you must view the PDF proof of the manuscript before you will be able to submit the revision. Note: if any parts of the electronic submission form have not been

completed, these will be noted by red message boxes - you will need to resolve these errors before you can submit the revision.

Author's Response to Decision Letter for (RSOS-201728.R4)

See Appendix G.

Decision letter (RSOS-201728.R5)

Dear Mr Keeble:

It is a pleasure to accept your Stage 2 Registered Report entitled "The evolution of coordination: A phylogenetic meta-analysis and systematic review" in its current form for publication in Royal Society Open Science.

Thank you for your fine contribution. On behalf of the Editors of Royal Society Open Science, we look forward to your continued contributions to the journal.

Kind regards,
Royal Society Open Science Editorial Office
Royal Society Open Science

on behalf of Professor Chris Chambers (Subject Editor)
openscience@royalsociety.org

Appendix A

We would first like to thank the editor for their recognition that this work has the potential for publication in Royal Society Open Science, and also the reviewers for their insightful comments regarding how this study can be improved. Given that such detailed comments were provided, and because we wanted to make sure we are replying to them in full, comments will be reproduced below with replies to comments indicated by bold font. Altered text in the manuscript has been highlighted, and reference to lines is made in this document where necessary.

1 Replies to reviewer 1

Although in principle I applaud the authors' goal of trying to bring together the results of many different cooperation studies with many different species, I am less optimistic this is going to bring forward the field of cooperation in comparative psychology for the reasons I expose below.

In particular, the study proposes the following hypotheses and questions:

(1) Does number of trials predict cooperative success in a delayed partner string-pull paradigm?

The delayed partner string-pull paradigm has produced positive results in a few species (but more species than the ones mentioned in this manuscript). However, this task is not a silver bullet with regards to what we can learn about animals's skills to cooperate. As it has been argued by other researchers (e.g. Seed & Jensen, 2011) there are several simple explanations that can explain how animals learn to pass these tests.

In light of this comment we have made the implications of this study clearer in the introduction, and changed terminology where appropriate from 'cooperative success' to 'coordination success', to avoid misunderstanding. We acknowledge more forcefully that there are several competing explanations for success in the string-pull task, and acknowledge that these may limit the conclusions of our study

[1] (see lines 42-44).

In addition, the number of trials is not the only variable that the different studies have changed. Different studies have also worked with different “delays” and different lengths of the rope, factors which can all influence whether or not and how quickly animals can succeed in this task. Therefore, I am afraid this will be a complicated analysis.

We acknowledge the concern of the reviewer that studies vary significantly in their design. Because of this concern, much detail has been added to the search protocol, study inclusion criteria, quality assessment and data extraction (sections 1.1, 1.2, 1.3, 1.4), and we hope that, because of this, the studies included in our analyses will be more specific to a task design. This will limit our conclusions, but hopefully make the conclusions we can draw more reliable. Furthermore, the aims of the study outlined in the introduction have been reframed so that one of our primary aims for the present study is to highlight the problems that variation in methods present for conducting the comparative meta-analyses necessary to make strong evolutionary conclusions, and thus provide an illustration of the concern the reviewer rightfully raises (lines 76-90, and section 1.3).

(2) Does inhibition predict cooperative success in a delayed partner string-pull paradigm? For the reasons explained above, there are actually not many species that have succeeded in the delay task under comparable conditions, so the N will be rather low and one would be comparing learning skills under very different conditions. In addition, the inhibition measure would be a species’ inhibition measure but not a measure obtained from the same populations that participated in the cooperation tasks. This could be acceptable if everything else was highly controlled. However, it is just another source of noise adding to the other ones, and when dealing with a rather low N.

This analysis has been removed from the study due to concerns expressed by both reviewers. We acknowledge that the data used in the analysis is likely to unreliable to provide any reliable conclu-

sions. Furthermore, as reviewer 1 points out, inhibition measures and measures of success in a coordination task would have been from different populations. The best way to assess this question would likely be to conduct further coordination experiments and including a measurement of individual inhibition, before comparing the relationship between individual inhibition and success in the coordination task across species. At the moment, the data is not reliable enough to assess this question [2].

(3) Do social dynamics (affiliation and tolerance) predict cooperative success? This is possibly the question I am more positive about, just because there are more studies that have looked at tolerance and affiliation. Nevertheless, I would emphasise that cooperation success, as it was measured in many of the studies, does not equate with an understanding of the contingencies and role of the partner in the cooperative endeavour. In other words, one would be looking at something about dyads' relationships and capacity to interact manipulating together food sources but not *collaborative skills* per se.

(4) Do brain size, social group size, or dietary factors predict cooperative understanding? As mentioned above, success in the delay task alone does not mean cooperative understanding. The operationalization of cooperative understanding is more complicated. This study proposal mentions gaze-following too, but that also is not necessarily a good dependent measure: we know that subordinates are often nervous in the presence of dominants when manipulating food rewards, which could lead to more "monitoring" behaviour.

The aims of the final two analyses have been reframed in terms of 'ability to coordinate in a delay string-pull task' as opposed to 'understanding of cooperative problem', given that the latter terminology may be misleading (see lines 95-100, and terminology changes throughout). As reviewer 1 has previously pointed out, there are competing explanations for success in the string-pull task, and given that subjects could still be successful in this task despite not fully 'understanding' the task, whatever that may mean, we agree that

framing our questions and measure in terms of ‘understanding’ was inappropriate. We hope that changing our terminology will, in this case, reframe the analysis in terms of the relationship between social dynamics and success in a task, or the relationship between certain evolutionary predictors and success in a task, without making any claims regarding whether or not animals are ‘understanding’ the task and all its intricacies. Gaze following has also been removed as a measure of ‘understanding’, as this could be a measure of many different things.

Other comments:

- In the pilot data presented, I noted that Chalmeau 1994 and Suchak et al. 2014 are two of the studies analysed but those studies did not use the same string-pulling task mentioned in the introduction as necessary to facilitate any conclusions about cooperation understanding. The apparatus of those studies were different to the one used in all the delay tasks, so although this may be acceptable, one needs to acknowledge that the contingencies for success with that apparatus are different ones. Individuals can pull alone without losing the opportunity to access the rewards, so waiting in that task is not necessarily needed. If the understanding measure used is “monitoring” behaviour, I am sceptical about it for the reasons mentioned above.

The authors hope that a more developed search protocol (section 1.1), and a more thorough inclusion protocol (sections 1.2 and 1.3) based on the PRISMA guidelines and diagram [3] will remove studies like those mentioned by reviewer 1 that are not as relevant to the questions or measures outlined in the present study proposal, but were included in pilot data due to a less refined protocol for inclusion. Furthermore, as mentioned above, measures of ‘understanding’ have been reframed as measures of ‘success in a coordination task’, to avoid misunderstanding.

- How informative is the correlation - number of trials and cooperation success? Certainly, with more experience cooperative partners will become more coordinated, but (1) are there significant differences in how quickly different species reach success? (2) Are the levels of success comparable across species? i.e. some species quickly being able to wait for up to 25 sec for the partners whereas others the maximum they wait is 5sec? (3) how is success being measured in those studies using different pulling apparatuses? Because co-acting is not necessarily cooperating.

The data extraction protocol has been amended so that variables measuring trials to success, delay time used in delay-release task, and whether the task used is delay- or simultaneous- release task are recorded from studies where possible (see lines 227-235). However, data may not be available from studies to conduct any statistical analysis on such data. As a result, a new aim of the present study will be to provide a comprehensive dataset that allow researchers to access and compare these aspects of string-pull studies across species (see lines 76-90). We hope that this will help researchers design future studies that are directly comparable to existing results, as well as illustrate the difficulties encountered when methods are used with slight variations.

2 Replies to reviewer 2

Comments to the Author(s) Keeble, Wallenberg and Price propose to meta-analyses and conduct a phylogenetic analysis of data from the co-operative string-pulling literature. I think the proposal is exactly the type of approach the field needs to explore more regarding evidence synthesis, and I think it's great that they've chosen to submit this as a registered report. However, as it stands, I think parts the protocol needs much greater specification, and there are several problems the researchers are likely to face when conducting this

meta-analysis which might prevent it from providing strong answers to their questions. Nevertheless, I see a large amount of merit in the proposal. I think that if the authors reframed it more in terms of exploring some of the difficulties in performing comparative meta-analyses, as well as focusing on the substantive questions about co-operation, then it could make a very nice RR. I've framed my comments around each of the proposed analyses:

We thank reviewer 2 for their encouraging comments. We also acknowledge, and are grateful for, the insight that this registered report provides an opportunity for us to explore the difficulties that the application of meta-analysis to the field presents, and an opportunity to create a valuable data resource for other interested researchers. As such we have attempted to reframe this registered report as a valuable exploration of the difficulties of research synthesis in comparative cognition, while simultaneously allowing us to collate multiple aspects of studies that will hopefully illuminate gaps in the literature and help direct future research, and assess their rigour using systematic review.

1. Does number of trials predict cooperative success in a delayed partner string-pull paradigm?

Could the authors specify more about how they will define the number of trials when extracting data, and similarly how they will define success on the string-pulling paradigm. I can see the definition they provide in the caption to Figure 1, but this could be detailed in text.

With trial data, how will the authors deal with differing numbers of training trials and testing trials overall – i.e., if they define the number of trials as the number of testing trials, what happens if studies have markedly different numbers of training trials – or if this information is not available? Similarly, how will the authors address cases where animals are tested sequentially with different partners, potentially across multiple studies, some of which may not have been published?

Differences between testing and training trials will be assessed

using quality assessment during systematic review (see section 1.3 and 1.4). For meta analysis, estimates assessing the relationships between number of trials and coordination success will only be included if that measure was obtained using overall number of trials (both training and test) a subject received (see lines 161-164).

Overall, I think the exact extraction procedure could be specified in greater detail, and the procedure might need to be expanded such that the authors can document the full extent of heterogeneity between the studies. Understanding this heterogeneity is key to understanding what the output of the meta-analyses means – the heterogeneity might be so large that the aggregate point estimates/CIs don't have much meaning – and I imagine most people would have high priors that that more trials leads to greater success anyway. I still think there is a lot of merit in collecting and summarizing this data, and the meta-analysis workflow will help this, but I think the main benefit of this proposal is in collecting and presenting information about individual study results and the between-study heterogeneity.

We hope that, by including a quality assessment stage (section 1.3) in this systematic review, we can include a critical summary of individual study results as a major part of the results section of this review. Furthermore, we have elaborated on analysing not only all studies together, but also include subgroup analyses of species, in an attempt to summarise not only findings and heterogeneity across and between all studies all studies, but also analyse findings and heterogeneity between and within species (see lines 283-302).

I have further questions about the models the authors are using for the meta-analysis, and how they will deal with repeated data from individuals (e.g., in the pilot report “Peron et al. 2011 Grey parrots” appears with three separate estimates – if I've found the paper correctly this comes from 3 parrots tested in in 3 consecutive experiments).

Data from studies conducting analysis at individual and group levels will be analysed both separately and together, and where indi-

viduals have been tested repeatedly, their estimates will be averaged across those tests (See lines 275-281).

As it stands, I don't think the current meta-analysis model accounts for phylogenetic relatedness – they might want to consider fitting a multi-level model which incorporates relatedness in (see e.g. Dougherty & Guillette, 2018 as an example), unless there is a reason why they would prefer to not include this information, but this should be explained.

If the present article goes through a revision, it would be nice if the pilot data, extraction protocol and code could be made available to help understand the procedures and models better.

Pilot data and code for both pilot analyses and proposed final analysis have been included for review. In light of reviewer 2's comments, models have been made multi-level to account for phylogenetic relatedness and also data from studies that appear repeatedly in the dataset (see lines 266-275).

2. Does inhibition predict cooperative success in a delayed partner string-pull paradigm?

This analysis proposes to see if inhibition scores from MacLean et al.'s study predicts success in the string-pulling. I have strong reservations about the ability of this analysis to have the statistical power to produce a meaningful analysis, most of all because the between-site replicability of the MacLean et al. data is largely untested, and where it has been tested it appears low – i.e., what have been billed as species differences in for example the cylinder task performance may not be so, (see Figure 3 and the following discussion in Farrar et al., 2020). Because the replicability of inhibition data may be low (as well as for the string pulling data) – this analysis might end up predicting noise with noise. I'm not completely opposed to the analysis being performed, but I think it should be heavily caveated if so. If the authors do decide to proceed, I'd encourage them to search further for inhibition data as there are many studies that use the same tasks as MacLean et al., to ensure they are getting as much data as possible to inform the analysis.

I'm also uncertain about what the best way to interpret the results of this analysis are. On the one hand, correlation does not mean causation, and there are likely many reasons why data from the inhibition task will correlate with data from the string pulling task, without there being a causal relationship between them. On the other hand, I think again it's incredibly likely that inhibition is causally related to some aspects of passing the string pulling task. If the authors did not find a positive result then I'd be pretty confident that this would be a false negative (i.e., some relationship between inhibition and string-pulling performance is very likely to exist), and if they do find the positive effect, I'm unsure whether the aggregated numerical estimates would mean very much, because of the heterogeneity between studies.

This analysis has been removed from the present study. Both reviewers rightly point out that using scores at the species level is inappropriate given that there is likely a lot of variation between individuals, and that the potential unreliability of the measures from the MacLean et al. [4] study means the analysis may be, at this point in time, largely redundant and too difficult to interpret.

3. Do social dynamics (affiliation and tolerance) predict cooperative success

4. Do brain size, social group size, or dietary factors predict cooperative understanding?

I have similar reservations about these analyses to the previous two. Again, I think it's highly likely that social dynamics will to some extent predict performance on a social co-operation task (with both causal and non-causal relationships), but it is also possible that the data quality/quantity are not there to perform meaningful quantitative analyses on these. I think the analyses will be useful, but interpreting their outputs may be difficult from a theoretical perspective.

When constructing the phylogenetic trees, how will the authors quantify the uncertainty in the output? They will be able to generate a tree that is the best fit according to some criteria, but if the overall data quality are poor, the tree might be quite likely to be inaccurate, i.e., there is no error control.

We recognise that the analyses proposed in this paper are likely to be difficult to interpret based on concerns of both reviewers. One primary cause of this is likely high variability in study designs. However, we hope that the now extended quality assessment protocol (section 1.3) will allow us to discuss where issues arise with comparability across studies and demonstrate the difficulties of research synthesis in comparative cognition, as well as help us include only the most comparable estimates in meta analysis. To assess the reliability of phylogenetic trees, the R package phangorn [5] will be used. Trees will be compared against each other, and the tree with the lowest AIC will be applied in phylogenetic generalised least squares models (see lines 246-262).

My last comments are not specific to a particular analysis:

5. The overall aim of the study

The proposal aims to make “conclusions about the evolutionary origins of cooperative problem solving” by analysing data from the co-operative string-pulling task. However, the generalizability of results from the string-pulling task to other tests (real or hypothetical) of co-operative behaviour is relatively unknown, and as such framing a meta/phylogenetic analysis of string-pulling data as an analysis of co-operative behaviour in general is excessive, in my view. The data will be relevant to this question, but not strong. It could instead be framed more narrowly around the string-pulling task.

Both reviewers rightly acknowledge that our claims about the conclusions that could be drawn from the proposed analysis are excessive. In response, we have narrowed our proposed aims and potential conclusions to success in a coordination task, and removed claims previously made that our analysis can tell us much about ‘understanding and problem-solving’ capabilities. Especially given that success in a coordination task may be a result of factors other than social cognition and/or ‘understanding’ [1], and given that ‘understanding’ and ‘problem solving’ are ill defined concepts in themselves. As such, we

have replaced these phrases with ‘coordination success’ and ‘coordination ability’ where relevant.

6. The search protocol and quality control

The search protocol and extraction procedure could be specified to a much a greater extent. One of the strongest outcomes of this study is a high-quality data resource on the string-pulling task. It’s important then to ensure that the search and extraction process are comprehensive and high quality. I’d recommend the authors to follow the PRISMA guidelines for reporting, and to add in several quality control stages to their extraction procedure. Regarding the search terms: searching for comparative data can be very difficult due to mass heterogeneity in how researchers report studies, and I think the Google scholar search and citation-based search will identify most studies. However, it would be good to have some verification stages here to check that key studies are not missing. It would be good if the search structure was specified more, too – for example, are the searches simply performed by inputting the four statements into google scholar as they are, or by using truncation and wildcards (e.g. searching for “cooperat*”), will they search for different combinations and spellings (cooperation as well as co-operation). It’s possible that the authors have already considered this but it would be great if the full structure of the search could be given. The authors may wish to complement their very general search of the entirety Google scholar with more targeted searches of specialist animal behaviour journals as a method of checking that all relevant publications have been identified.

The search protocol has been specified in greater detail, including truncated terms and a repetition of the search protocol conducted in specialist animal behaviour/cognition journals. Furthermore, a quality assessment stage has been included to address the problem of differences in methods and methodological rigour across studies (see sections 1.1 and 1.3).

Could the inclusion criteria be specified further still – will the authors include conference abstracts/thesis chapters/non-published articles? I think they definitely should, but they might want to flag these studies when examining

publication bias.

More detail has been added to the inclusion criteria (sections 1.1, 1.2). The study will include theses and abstracts, and will also approach principal investigators of existing studies to attempt to ascertain unpublished data and results.

In the extraction process, it is unclear what exactly is being extracted and how. It would be great to build a more comprehensive search protocol, and in particular include details on the study design (training trials, sample sizes, sequential testing etc.) and identify studies in which the participants may overlap. I think it is important to have some form of quality control in this process too, either through double extracting a decent percentage of the studies to assess inter-extractor agreement, or more preferable, to have each extraction checked by another researcher.

Data extraction will be conducted twice by the primary author of the study, with a week separating the two extraction phases. 10% of extractions will be repeated by another author on the paper to assess inter-extractor reliability. Details on study design have been included in the data extraction protocol (see lines 228-235), in an attempt to facilitate the aim of creating a useful data source for other researchers on the subject of coordination tasks.

7. Detecting publication bias

I think it's great the authors are paying attention to publication bias in the analysis, and I think the approach could be expanded to examine publication bias in some of the other data they collect too, e.g., by examining the p-value distributions of reported correlations between tolerance and success on the string pulling task. Currently, their interpretation of the funnel plot and significant Egger's regression test is possibly excessive – a statistically significant asymmetry does not “demonstrate [as in prove] that there are several studies missing”, although it is compatible with that (see e.g. Sterne et al., 2011). There are reasons why comparative datasets from across many species may have asymmetric funnels – for example what if the smaller sample studies - with larger associ-

ated errors – are performed on species that are disproportionately likely to pass the test (e.g. a small number of elephants or chimpanzees), whereas the larger sample studies are performed on species that might be less likely to pass the test (e.g. dogs or monkeys). I don't necessarily agree with this reasoning, and think the funnel test is good evidence of publication bias, but currently I believe its overinterpreted. I think it's incredibly difficult to assess publication bias in comparative datasets because of the massive heterogeneity in the laboratories and species publishing and performing the tasks. I'd encourage the authors to examine subgroups (e.g. by colour coding their funnel plots by species/groups), particularly if they get a decent amount of data on any one particular group. Finally, the authors may wish to consider alternative methods of detecting publication bias – for example by identifying thesis chapters or conference abstracts that have not been published, or even by simply asking researchers in the field if they have performed any string-pulling tasks that have gone unpublished. Some information like this would complement the statistical analysis massively, in my opinion.

We would like to thank reviewer 2 for acknowledging and encouraging our attempt to assess publication bias in this area of research. We have added further tests of publication bias (Orwin's fail safe N, an assessment of p-value distribution, cumulative meta-analysis ordered by sample size, and a meta-regression model assessing differences in findings between published and unpublished data if possible). Furthermore, these tests will be repeated with species subgroups should sufficient data be available to do so. See lines 312-334.

Overall, I think this will be a really valuable study. I think the first aim should be in creating a well-document data resource around the string pulling task, from which they will be able to assess how much information the meta-analyses and phylogenetic analysis can provide. There are several key problems to navigate for the analyses (heterogeneity, structure of the models, data quality) and it's tricky to review this without having much of the information about

the studies yet. This will only be available after extractions have been completed, and so I think it is really important if IPA is to be given to know the extraction process will be very high quality, and that a critical assessment of the strengths of the meta-analyses will be performed after this, or to have another review/discussion about the proposed analyses after extractions have been completed. If it turns out that the data are too low in quality (or in quantity, or homogeneity) to perform strong analyses, then this information is still very valuable to the field, and the process by which the authors will decide this will be a useful methodological advance in the field.

We thank reviewer 2 for their insightful suggestions. We have attempted to reframe this report so that the collection and quality assessment of coordination studies, and data from these studies, are a primary aim. We hope that the remaining analyses will serve as a demonstration of the difficulties that research synthesis of comparative data presents, but also provide a useful demonstration of the opportunities available to comparative researchers should more reliable data be available/collected.

References

- [1] A. M. Seed, K. Jensen, Large-scale cooperation, *Nature* 472 (7344) (2011) 424–425.
- [2] B. Farrar, K. Voudouris, N. Clayton, Replications, comparisons, sampling and the problem of representativeness in animal behavior and cognition research.
- [3] D. Moher, A. Liberati, J. Tetzlaff, D. G. Altman, P. Group, et al., Preferred reporting items for systematic reviews and meta-analyses: the prisma statement, *PLoS med* 6 (7) (2009) e1000097.
- [4] E. L. MacLean, B. Hare, C. L. Nunn, E. Addessi, F. Amici, R. C. Anderson, F. Aureli, J. M. Baker, A. E. Bania, A. M. Barnard, et al., The evolution

of self-control, *Proceedings of the National Academy of Sciences* 111 (20)
(2014) E2140–E2148.

- [5] K. P. Schliep, *phangorn: phylogenetic analysis in r*, *Bioinformatics* 27 (4)
(2011) 592–593.

Appendix B

Comments from the editor and reviewers have been reproduced here so that we can attempt to address them all in a thorough manner. Our responses are indicated in bold.

1 Associate Editor Comments to Author (Professor Chris Chambers)

The revised manuscript was returned to the two original reviewers. Both reviewers offer a range of comments on the updated proposal. Reviewer 2 is broadly positive but also notes several methodological issues requiring further clarification. As in the first round, Reviewer 1 is again the more critical of the reviewers, questioning the overarching rationale for the research and the heterogeneity of the studies included in the meta-analysis. These are major issues that will need to be thoroughly addressed to achieve IPA.

Given the progress made in improving the design, and the enthusiasm of Reviewer 2, I want to give the authors a final opportunity to address these concerns before issuing a final Stage 1 editorial decision.

We are grateful to the editor for their further consideration of the present registered report, and will aim to update the manuscript in light of further comments from reviewers. We would also like to express our thanks to both reviewers for their very useful comments and suggestions, and we hope we have addressed them appropriately.

2 Reviewer: 1

Although this revised version addresses some of the issues I raised in the previous round, I still remain unconvinced that a phylogenetic meta-analysis in this field of research can currently bring the field forward. The reasons are, as I commented before, the following:

- There are very few species which have passed the delay task (i.e. chim-

panzees, elephants, maybe dogs and dolphins), so with such scarce evidence, I am not sure what is the point of a phylogenetic analysis. In addition, it is now suggested to constrain more which studies enter the analysis to make the comparison more valid (due to the many different testing and experimental protocols), but how many studies would we be left with? if currently only 2-3 species have been considered to pass the task?

- **In light of comments from both reviewer 1 and reviewer 2, we have decided to no longer require studies to adhere to certain quality assessment criteria for their inclusion in statistical analyses. As both reviewer 1 and 2 have noted, doing so would likely have resulted in too few studies being included in such an analysis. Furthermore, we hope that our revised introduction and rationale demonstrate that the systematic review (especially the quality assessment) aspect of the present study and the actual collation of results from animal coordination studies is a primary aim (see lines 78-156). There may indeed be too few studies and findings to conduct a thoroughly reliable phylogenetic analysis after data synthesis (in which case, such an analysis will not be conducted), but we hope that the data synthesis itself, and the assessment of the quality of the studies that can be included, will be a useful resource for the field in directing researchers towards conducting studies that are necessary for asking phylogenetic questions. The feasibility of phylogenetic analyses will be assessed post data collection, and will not be conducted if certain constraints are not met (see lines 405-409 and 417-420). However, we believe that starting to synthesise data towards this end is still a valuable pursuit for the field.**
- In this revised version Cooperation has been substituted with "coordination", but evidence for coordination can only come from studies showing

active efforts to synchronise actions in time and space. Therefore, many studies using other cooperation apparatuses are less indicative of coordination (Suchak et al. Mendres and De Waal, 2003..) because subjects can pull indiscriminately until the apparatus moves (co-occurring when another subject has joined). However, if I understand correctly, these different studies would also enter the analysis. I don't think by changing the word to "coordination", authors can circumvent the difficulty of making conclusions about the mechanisms that bring success about in these co-acting tasks.

- **We hope that a clarification of rationale in the introduction demonstrates that making conclusions about mechanisms is not a primary goal of the present research. Rather, the primary goal is to assess the plausability and validity of evidence synthesis in the field of comparative cognition/psychology (see line 121-156). Furthermore, inclusion criteria has been amended so that only studies using apparatus where the task is failed if only one individual pulls are included, which we hope addresses a large source of heterogeneity between studies that may be included in analysis (see lines 201-204). We hope this addresses reviewer 1's comment that our study would not have extracted data from coordination paradigms only.**

- The question about the quality of the relationship between subjects and success in a cooperation task is interesting, but it has already been shown experimentally in several species (e.g. Capuchin monkeys, De Waal & Davis, 2003; Tonquean macaques, Petit et al. 1992; Chimpanzees, Melis et al. 2006; Rooks, Seed et al. 2008). These previous studies have shown the effect of the relationship between partners on cooperative problem solving behaviour *within* a species and *across* species.

The introduction has been amended to acknowledge this point from Reviewer 1 (see lines 36-45). However, we also note that

such results are yet to be standardised and compared, which would allow more reliable comparisons of effect sizes across species [1] and can most effectively be achieved using meta analytic methods (see lines 42-45).

One last point is that there is a probably not a *single* method or experimental protocol that can best investigate animals' capacity to solve problems cooperatively with others (and coordinate actions). The type of tasks that can be used with primates are maybe not equally suited for birds or dolphins and the other way around. Therefore, the delay task with the string-pulling paradigm should not be seen as the "silver bullet" to understanding Cooperation/Coordination, but instead as one task that together with others may give us some insight into animals' capacity to coordinate actions with each other.

What we urgently need are multiple paradigms and measures for each species and/or the exact same experiment across different species. Instead, what we currently have is one unique method implemented in myriad ways.

We agree with reviewer 1's point that no single task can provide definitive answers in any sub-field of the evolution of cognition. However, we believe that certain tasks do provide a useful unit of comparison in a field that must use comparisons when answering its central questions. This is especially so when potentially comparable variations on a task have been used (see lines 198-201). The comparability of such variations must and can be assessed using systematic review. Thus we argue that our study could demonstrate a useful approach towards gathering evidence for answering central questions in the field of comparative cognition, whilst also acknowledging that future evidence synthesis may need to be conducted along dimensions different to those chosen by us for the present study (see lines 86-106).

3 Reviewer: 2

The revised proposal has addressed many of the previous comments, increased the rigour of the proposal and added in several additional analyses. This will be a complicated project, and there is still work to be done to ensure it would be effective should IPA be given, but I still think it is a useful project.

3.1 Structure

The proposal would be easier to follow if the introduction was split in two, with a first section outlining the original research goals (meta-analysing string pulling data), and then a section outlining the challenges of doing so, rather than having the challenges spread throughout the introduction. This may be a personal preference, but I think it would improve the article's usability as a resource for meta-analysis in the area, too.

We agree with reviewer 2 that a restructuring of the introduction is necessary following the changes made in the first round of peer review, and we hope that we have clarified the rationale of the study by doing so (see lines 1-156.

3.2 Quality control and assessment

The authors have increased the amount of quality control and assessment stages in the revision.

a) Searches Although the searches are not fully comprehensive (e.g., animal cognition journals such as The Journal of Comparative Psychology, International Journal of Comparative Psychology, Behavioural Processes, Learning and Behavior are not included in the journal search), there is enough redundancy across the three searches to ensure that the vast majority of relevant articles will be identified.

b) Extraction The extraction process, where the articles are extracted twice by the same individual and then another coding 10% seems fine, although if

there are inconsistencies across this 10% more papers will need double coding.

More papers will be double coded if inter-extractor reliability on the first 10% is low (see lines 292-293).

c) Quality Checklist The authors have introduced a quality checklist – a modified NICE quality appraisal checklist. I applaud the use of such a checklist, and think this is an interesting project in itself, so should be retained. However, using the checklist as a basis for inclusion/exclusion in the meta-analysis might be a problem, as I'm unsure how the authors will be able to judge some of the questions, and it is likely that most studies would end up being excluded if the checklist was followed strictly.

For example, for question 1b:

Do the selected participants or areas represent the eligible population or area? (e.g. were both sexes represented fairly equally? Were a variety of ages represented fairly equally?)

How will the authors know what the eligible populations are without contacting every study author to see which animals were available? Often in captivity the ages of the animals are similar, so many studies will not represent a variety of ages fairly.

For question 2b:

How well were likely confounding factors identified and controlled?

This would require the authors to pretty much peer-review each paper, unless the authors wanted to decide on a few key confounds for the string pulling task and check these in each paper.

Performing the checklist will be an interesting exercise, possibly to quantify just how far these studies are from the strong epidemiological studies that the quality appraisal form was designed to identify. However, I doubt it can be used effectively as a basis of inclusion/exclusion unless it heavily modified.

The checklist, although still an integral part of the overall study, has been abandoned as a method for the exclusion of studies in the proposed statistical analyses (see lines 280-286).

3.3 The reliability of the phylogenetic tree

AIC will be used to compare competing trees, but this will only compare the fit of the trees relative to each other, and not absolutely, i.e., the tree with the lowest AIC will not necessarily be a good tree, it will just be the best fitting of all the ones compared on the current dataset. I think this needs to be made clearer in the manuscript, particularly given the noise and limited amount of data that will be informing the trees.

We hope that this has now been made clearer in the manuscript (see lines 327-329).

3.4 Meta-analysis/statistical analyses

The meta-analytic model has been updated to be multi-level, which is good as it can attempt to account for phylogenetic relatedness. However, I don't think the current approach, which includes species and genus only (random = 1—Paper/Species/Genus), will capture most of this phylogenetic information – it would only be useful where many species come from the same genus.

A couple of examples of how to include phylogeny with the `rma.mv` function can be found in Cinar et al., 2020 (Model 9) and Davies et al., 2020. Both of these provide good descriptions of how they generate the phylogenetic tree (Davies et al. in particular for when distantly related taxa are used), and provide example code in the supporting information. The approach is to include a single random effect of phylogeny, based on a correlation matrix derived from a phylogenetic tree. Both Cinar et al. and Davies et al. use the same APE package to generate these as the current authors are using, so their code should be adaptable.

Meta analysis models have been amended to include matrix structures based on phylogenetic trees following Davies et al. [1] (see lines 344,382, and RegCode.R file)

The authors have a large number of analyses planned, and they could consider how they will control for multiple testing/whether they think it is an issue.

Multiple testing adjustments have been added into the statistical analysis to control for multiple testing (see lines 421-425, and Reg-Code.R).

3.5 Data and code

Thank-you for sharing the data and code. I managed to reproduce the pilot analysis fully and the data file was readable. When the full dataset is generated, it would be useful to have a data-dictionary to accompany the file explain precisely what each column is and how it was coded – as I expect this dataset might be of interest to many people.

We thank reviewer 2 for the suggestion of a data dictionary, and will produce one once the full dataset is generated.

Overall, the new proposal is improved, and I still think it is a feasible and worthwhile project. I share Reviewer 1's concerns about the ability of the project to provide strong answers to the original questions of interest. But even if it fails to answer these questions effectively, then highlighting the barriers around synthesizing evidence like would be a very useful paper. The feasibility of these analyses will depend on the quantity and quality of the overall dataset, which we will only know when it has been extracted. Therefore, it would be useful to conduct a feasibility assessment of the different analyses after the data have been extracted but before conducting them. If IPA is given to the project, I expect a longer Stage 2 review may be necessary than with other registered reports. I still think the project is very interesting and will be of interest those interested in string-pulling tasks but also those interested in evidence synthesis in the field.

We thank reviewer 2 for their interest in the present proposed project. We agree that the feasibility of statistical analyses will depend upon dataset quality, and we have added criteria for conducting certain analyses after dataset generation (see lines 405-409 and 417-420). We acknowledge that phylogenetic analyses may be difficult to

conduct given available information, and will assess the feasibility of these analyses post data collection. Furthermore, we hope that the larger role of quality assessment and data generation in our primary aims for the study will provide a useful resource for researchers in comparative cognition, regardless of the feasibility of these analyses at present. We hope we have represented this aim clearly in the manuscript (see lines 1-156).

3.6 Minor point

1. Code reproducibility Code, line 355 there is a comma after the first value in `ref = c("HM015213",)` This should be removed to read `ref = c("HM015213")` I was unable to run the code of the phylogenetic analysis – but I assume this just hasn't been completed yet.

We offer our apologies. The code for phylogenetic analysis is currently incomplete as information regarding the generated dataset will be necessary before some details can be added. It is currently serving as a template for those later details. This should have been made clear prior to submission and a note has been added to the code file.

References

- [1] A. D. Davies, Z. Lewis, L. R. Dougherty, A meta-analysis of factors influencing the strength of mate-choice copying in animals, *Behavioral Ecology* 31 (6) (2020) 1279–1290.

Appendix C

Once again we thank the editor and reviewers for their helpful comments and, this time, in-principle acceptance of our work.

We have addressed reviewer 2's comment by removing 'that researchers generalise findings from very small samples to a much larger population...' as a 'major issue' in the introduction (see lines 2-3), as reviewer 2 rightly points out that many areas of the field of Comparative Cognition do very well with small samples [1].

Finally, 'met' has been changed to 'assessed' on line 6 so as not to overstate the utility of systematic review and meta analytic methods in meeting the challenges stated in the opening of the introduction.

References

- [1] P. L. Smith, D. R. Little, Small is beautiful: In defense of the small-n design, *Psychonomic bulletin & review* 25 (6) (2018) 2083–2101.

Appendix D

Dear Professor Chris Chambers,

We thank you once again for the careful consideration of our work by both editors at Royal Society Open Science and external reviewers. We hope that we present here a piece of high quality research that has been of higher quality because of the registered report process.

Both the link to the pre-registration and final dataset can be found at the top of page 18 in the stage 2 manuscript. Both are pre-registration and data are held on the Open Science Framework.

No data other than pilot data was collected or analysed prior to the date of in principle acceptance.

We would be grateful for any feedback on the paper, and hope that it meets the rigorous scientific standards of your journal.

Sincerely,

Liam Keeble

Appendix E

The evolution of coordination: A phylogenetic meta-analysis and systematic review

Liam Keeble¹, Joel C. Wallenberg² and Elizabeth E. Price³

¹Henry Wellcome Building, Medical School, Newcastle Upon Tyne,
NE2 4HH, United Kingdom

²Percy Building, School of English Literature, Language and
Linguistics, Newcastle University, Newcastle upon Tyne, NE1
7RU, United Kingdom

³School of Psychology, Newcastle University

Keywords

Cooperation, Problem-solving, Meta-analysis, Coordination, Cognition

Corresponding author

Liam Keeble: L.Keeble1@newcastle.ac.uk

Abstract

[revised manuscript text omitted]

work: <https://osf.io/hr6ma/>.
**Declaration of interests**
The authors report no conflicts of interest regarding the authorship or publica-
tion of this article.
**Authors' contributions**
The conceptual and methodological details of the present study were developed
by all authors. LK carried out the search strategy, data extraction and sta-
tistical analyses. The results were written by LK. All authors contributed to
the discussion. An individual external to the study authors double coded a
proportion of papers collected for the study.
**Acknowledgements**
The authors would like to thank Ben Farrar and one other, anonymous reviewer
for their very helpful comments on study protocol and the paper itself. We
would also like to thank Robin Watson (RW) for double coding part of our
dataset.
**Funding**
This work was supported by the NINE Doctoral Training Partnership.
**References**
- [1] B. G. Farrar, D. M. Altschul, J. Fischer, J. van der Mescht, S. Placì, C. A.
Troisi, A. Vernouillet, N. S. Clayton, L. Ostojić, Trialling meta-research in
comparative cognition: Claims and statistical inference in animal physical
cognition, *Animal behavior and cognition* 7 (3) (2020) 419.
- [2] E. Fehr, U. Fischbacher, The nature of human altruism, *Nature* 425 (6960)
(2003) 785.
- [3] M. Tomasello, A. P. Melis, C. Tennie, E. Wyman, E. Herrmann, Two Key
Steps in the Evolution of Human Cooperation, *Current Anthropology* 53 (6)
(2012) 673–692. doi:10.1086/668207.
URL <http://www.journals.uchicago.edu/doi/10.1086/668207>
- [4] R. Boyd, P. J. Richerson, Culture and the evolution of human coopera-
tion, *Philosophical Transactions of the Royal Society B: Biological Sciences*
364 (1533) (2009) 3281–3288. doi:10.1098/rstb.2009.0134.
- [5] A. P. Melis, D. Semmann, How is human cooperation different?, *Philosophical*
*Transactions of the Royal Society B: Biological Sciences* 365 (1553)
(2010) 2663–2674. doi:10.1098/rstb.2010.0157.
- [6] T. Clutton-Brock, Cooperation between non-kin in animal societies, *Nature*
799 462 (7269) (2009) 51.

- [7] A. P. Melis, B. Hare, M. Tomasello, Engineering cooperation in chim-
panzees: tolerance constraints on cooperation, *Animal Behaviour* 72 (2)
(2006) 275–286. doi:10.1016/j.anbehav.2005.09.018.
- [8] S. Hirata, K. Fuwa, Chimpanzees (pan troglodytes) learn to act with other
individuals in a cooperative task, *Primates* 48 (1) (2007) 13–21.
- [9] K. Jaakkola, E. Guarino, K. Donegan, S. L. King, Bottlenose dolphins
can understand their partner’s role in a cooperative task, *Proceedings*
*of the Royal Society B: Biological Sciences* 285 (1887) (2018) 20180948.
doi:10.1098/RSPB.2018.0948.
URL [http://rspb.royalsocietypublishing.org/lookup/doi/10.](http://rspb.royalsocietypublishing.org/lookup/doi/10.1098/rspb.2018.0948)
[1098/rspb.2018.0948](http://rspb.royalsocietypublishing.org/lookup/doi/10.1098/rspb.2018.0948)
- [10] J. M. Plotnik, R. Lair, W. Suphachoksakun, F. B. M. de Waal, Elephants
know when they need a helping trunk in a cooperative task, *Proceedings*
*of the National Academy of Sciences* 108 (12) (2011) 5116–5121.
- [11] M. Heaney, R. D. Gray, A. H. Taylor, Keas perform similarly to chim-
panzees and elephants when solving collaborative tasks, *PLoS ONE* 12 (2)
(2017) 1–13. doi:10.1371/journal.pone.0169799.
- [12] A. M. Seed, N. S. Clayton, N. J. Emery, Cooperative problem solving in
rooks (*Corvus frugilegus*), *Proceedings of the Royal Society B: Biological*
*Sciences* 275 (1641) (2008) 1421–1429. doi:10.1098/rspb.2008.0111.
- [13] J. J. Massen, C. Ritter, T. Bugnyar, Tolerance and reward equity predict
cooperation in ravens (*Corvus corax*), *Scientific Reports* 5 (2015) 1–11.
doi:10.1038/srep15021.
URL <http://dx.doi.org/10.1038/srep15021>
- [14] F. Péron, L. Rat-Fischer, M. Lalot, L. Nagle, D. Bovet, Cooperative prob-
lem solving in African grey parrots (*Psittacus erithacus*), *Animal Cognition*
14 (4) (2011) 545–553. doi:10.1007/s10071-011-0389-2.
- [15] L. Ostojić, N. S. Clayton, Behavioural coordination of dogs in a coopera-
tive problem-solving task with a conspecific and a human partner, *Animal*
*Cognition* 17 (2) (2014) 445–459, **Academic journal search engine**.
doi:10.1007/s10071-013-0676-1.
- [16] B. Hare, A. P. Melis, V. Woods, S. Hastings, R. Wrangham, Tolerance Al-
lows Bonobos to Outperform Chimpanzees on a Cooperative Task, *Current*
*Biology* 17 (7) (2007) 619–623. doi:10.1016/j.cub.2007.02.040.
- [17] C. Tassin de Montaigu, K. Durdevic, D. Brucks, A. Krasheninnikova,
835 A. von Bayern, Blue-throated macaws (*ara glaucogularis*) succeed in a co-
836 operative task without coordinating their actions, *Ethology* 126 (2) (2020)
267–277.
- [18] K. Asakawa-Haas, M. Schiestl, T. Bugnyar, J. J. Massen, Partner choice
in raven (*corvus corax*) cooperation, *PLoS ONE* 11 (6) (2016) 1–15. doi:
10.1371/journal.pone.0156962.
- [19] S. Marshall-Pescini, C. Basin, F. Range, A task-experienced partner does
not help dogs be as successful as wolves in a cooperative string-pulling task,
*Scientific reports* 8 (1) (2018) 1–12.
- [20] M. Suchak, T. M. Eppley, M. W. Campbell, F. B. de Waal, Ape duos and
trios: spontaneous cooperation with free partner choice in chimpanzees,
*PeerJ* 2 (2014) e417. doi:10.7717/peerj.417.
URL <https://peerj.com/articles/417>
- [21] M. Suchak, J. Watzek, L. F. Quarles, F. B. de Waal, Novice chimpanzees
cooperate successfully in the presence of experts, but may have limited
understanding of the task, *Animal Cognition* 21 (1) (2018) 87–98. doi:
10.1007/s10071-017-1142-2.
- [22] A. D. Davies, Z. Lewis, L. R. Dougherty, A meta-analysis of factors influ-
encing the strength of mate-choice copying in animals, *Behavioral Ecology*
854 31 (6) (2020) 1279–1290.

- [23] A. M. Seed, K. Jensen, Large-scale cooperation, *Nature* 472 (7344) (2011)
424–425.
- [24] L. Ostojic, The evolution of social cognition: the case of animal cooperative
problem solving (2020).
- [25] J. C. Valentine, Judging the quality of primary research, *The handbook of*
*research synthesis and meta-analysis 2* (2009) 129–146.
- [26] W. Wood, A. H. Eagly, Advantages of certainty and uncertainty, *The Hand-*
*book of Research Synthesis and Meta-Analysis* (2009) 455.
- [27] M. P. Crawford, The cooperative solving of problems by young chim-
panzees, *Comp Psychol Monogr* 14 (1937) 1–88.
- [28] D. Moher, A. Liberati, J. Tetzlaff, D. G. Altman, P. Group, et al., Pre-
ferred reporting items for systematic reviews and meta-analyses: the prisma
statement, *PLoS med* 6 (7) (2009) e1000097.
- [29] M. Borenstein, L. V. Hedges, J. P. Higgins, H. R. Rothstein, *Introduction*
*to meta-analysis*, John Wiley & Sons, 2011.
- [30] B. Farrar, K. Voudouris, N. Clayton, Replications, comparisons, sampling
and the problem of representativeness in animal behavior and cognition
research.
- [31] N. Borrego, Socially tolerant lions (*panthera leo*) solve a novel cooperative
problem, *Animal Cognition* 23 (2) (2020) 327–336.
- [32] N. I. for Health, C. E. G. Britain), *Methods for the development of NICE*
*public health guidance*, National Institute for Health and Care Excellence,
2012.
- [33] J. Cohen, A coefficient of agreement for nominal scales, *Educational and*
*psychological measurement* 20 (1) (1960) 37–46.
- [34] A. Del Re, M. A. Del Re, Package ‘compute. es’ (2012).
- [35] E. Paradis, K. Schliep, *ape* 5.0: an environment for modern phylogenetics
and evolutionary analyses in *r*, *Bioinformatics* 35 (3) (2019) 526–528.
- [36] K. P. Schliep, *phangorn*: phylogenetic analysis in *r*, *Bioinformatics* 27 (4)
(2011) 592–593.
- [37] D. A. Benson, M. Cavanaugh, K. Clark, I. Karsch-Mizrachi, D. J. Lipman,
886 J. Ostell, E. W. Sayers, Genbank, *Nucleic acids research* 41 (D1) (2012)
D36–D42.
- [38] R. C. Edgar, *Muscle*: multiple sequence alignment with high accuracy and
high throughput, *Nucleic acids research* 32 (5) (2004) 1792–1797.
- [39] C. Heibl, *Phyloch*: R language tree plotting tools and interfaces to di-
verse phylogenetic software packages, Available online at: [http://www.](http://www.christophheibl.de/Rpackages.html)
[christophheibl.de/Rpackages.html](http://www.christophheibl.de/Rpackages.html) (2008).
- [40] O. Gascuel, *Bionj*: an improved version of the *nj* algorithm based on a
simple model of sequence data., *Molecular biology and evolution* 14 (7)
(1997) 685–695.
- [41] K. P. Burnham, D. R. Anderson, K. P. Huyvaert, Aic model selection and
multimodel inference in behavioral ecology: some background, observa-
tions, and comparisons, *Behavioral ecology and sociobiology* 65 (1) (2011)
23–35.
- [42] E.-J. Wagenmakers, S. Farrell, Aic model selection using akaike weights,
*Psychonomic bulletin & review* 11 (1) (2004) 192–196.
- [43] R. Freckleton, P. Harvey, M. Pagel, Phylogenetic dependence and ecological
data: a test and review of evidence, *Am. Nat* 160 (2002) 716–726.
- [44] M. Pagel, Inferring the historical patterns of biological evolution, *Nature*
905 401 (6756) (1999) 877–884.
- [45] A. N. Iwaniuk, J. E. Nelson, Can endocranial volume be used as an estimate
of brain size in birds?, *Canadian Journal of Zoology* 80 (1) (2002) 16–23.
- [46] E. L. MacLean, B. Hare, C. L. Nunn, E. Addessi, F. Amici, R. C. Anderson,
F. Aureli, J. M. Baker, A. E. Bania, A. M. Barnard, et al., The evolution
of self-control, *Proceedings of the National Academy of Sciences* 111 (20)
(2014) E2140–E2148.
- [47] A. R. DeCasien, S. A. Williams, J. P. Higham, Primate brain size is pre-
dicted by diet but not sociality, *Nature ecology & evolution* 1 (5) (2017)
1–7.
- [48] M. W. Colbert, R. Racicot, T. Rowe, Anatomy of the cranial endocast
of the bottlenose dolphin, *tursiops truncatus*, based on hrxt, *Journal of*
*Mammalian Evolution* 12 (1-2) (2005) 195–207.
- [49] W. Viechtbauer, Conducting meta-analyses in r with the metafor package,
*Journal of statistical software* 36 (3) (2010) 1–48.
- [50] L. R. Dougherty, L. M. Guillette, Linking personality and cognition: a
meta-analysis, *Philosophical Transactions of the Royal Society B: Biological*
*Sciences* 373 (1756) (2018) 20170282.
- [51] R. G. Orwin, A fail-safe n for effect size in meta-analysis, *Journal of edu-*
*cational statistics* 8 (2) (1983) 157–159.
- [52] J. Pinheiro, D. Bates, S. DebRoy, D. Sarkar, S. Heisterkamp, B. Van Willi-
gen, R. Maintainer, Package ‘nlme’, Linear and nonlinear mixed effects
models, version 3 (1) (2017).
- [53] S. Holm, A simple sequentially rejective multiple test procedure, *Scandi-*
*navian journal of statistics* (1979) 65–70.
- [54] M. Primate, D. M. Altschul, M. J. Beran, M. Bohn, J. Call, S. DeTroy,
S. J. Duguid, C. L. Egelkamp, C. Fichtel, J. Fischer, et al., Establishing an
infrastructure for collaboration in primate cognition research, *PLoS One*
14 (10) (2019) e0223675.
- [55] P. Bateson, K. N. Laland, Tinbergen's four questions: an appreciation and
an update, *Trends in ecology & evolution* 28 (12) (2013) 712–718.
- [56] P. L. Smith, D. R. Little, Small is beautiful: In defense of the small-n
design, *Psychonomic bulletin & review* 25 (6) (2018) 2083–2101.
- [57] B. Farrar, L. Ostojic, The illusion of science in comparative cognition
(2019).
- [58] J. S. Martin, S. Koski, T. Bugnyar, A. V. Jaeggi, J. J. Massen, Proso-
ciality, social tolerance and partner choice facilitate mutually beneficial
cooperation in common marmosets, *callithrix jacchus*, *Animal Behaviour*
173 (2021) 115–136.
- [59] E. L. MacLean, L. J. Matthews, B. A. Hare, C. L. Nunn, R. C. Anderson,
F. Aureli, E. M. Brannon, J. Call, C. M. Drea, N. J. Emery, et al., How does
cognition evolve? phylogenetic comparative psychology, *Animal cognition*
947 15 (2) (2012) 223–238.
- [60] S. J. Shettleworth, The evolution of comparative cognition: is the snark
still a boojum?, *Behavioural processes* 80 (3) (2009) 210–217.
- [61] F. A. Beach, The snark was a boojum., *American Psychologist* 5 (4) (1950)
115.
- [62] A. Krasheninnikova, P. K. Y. Chow, A. M. von Bayern, Comparative cog-
nition: Practical shortcomings and some potential ways forward., *Canadi-
an Journal of Experimental Psychology/Revue canadienne de psychologie
expérimentale* 74 (3) (2020) 160.

Appendix F

Data Dictionary

Liam Keeble, Joel Wallenberg, Bess Price

Emboldened variables are to be double coded and analysed for agreement. If any variables are not presented in paper, code as "NA".

- Title: Title of publication.
- Authors: Authors of publication.
- Year: Year of publication.
- Removed: Removed during assessing studies against inclusion criteria (Yes/No). (Categorical variable)
- Reason: Reason for removal during assessing studies against inclusion criteria.(Description)
- Latin: The latin name of the species in study.
- Code: GenBank code identifier for species genetic information. (Categorical variable)
- ExperienceEst: Estimate for the relationship between success in a simultaneous release task and number of trials/attempts at the task (Correlation).
- ExperienceVar: Variance for experience estimate for the relationship between success in task and number of trials/attempts at task.
- **OrigEst0**: Description of estimate taken directly from a paper for the relationship between success in a simultaneous release task and number of trials/attempts at the task from paper before conversion (e.g."correlation coefficient of 0.7"). (Description)
- **DelayPValues**: P value reported for a statistical test on success in a delay task. (e.g. "p=0.0012") (Numeric)
- **PDescription**: Description of what p-value is representing in study. (e.g."Difference between groups", or "Performed better than chance"). This is often unclear in studies. In which case, it can be described as such (e.g."unclear"). (Description)
- DelayEst: Estimate of success for delay coordination task (rate of success). (Numerical variable: percentage/fraction)

- **DelayVar:** Variance for estimate of success for delay coordination task.
- **OrigEst1:** Description of estimate taken directly from paper for success rate in a delay task, where a participant's partner has been held back from the apparatus (e.g. "50% success rate", or "Dyads were successful 10 out of 20 trials on average"). (Description)
- **TolEst:** Estimate for the relationship between success in a simultaneous release and the tolerance score of a dyad (Correlation).
- **TolVar:** Variance for tolerance estimate for the relationship between success in a simultaneous release task and the tolerance score of a dyad.
- **OrigEst2:** Description of estimate taken directly from paper for relationship between dyad tolerance and coordination success (e.g. "Correlation coefficient of 0.5"). (Description)
- **AffEst:** Estimate for the relationship between success in a simultaneous release and the affiliation score of a dyad. (Correlation)
- **AffVar:** Variance for tolerance estimate for the relationship between success in a simultaneous release task and the affiliation score of a dyad.
- **OrigEst3:** Description of estimate taken directly from paper for relationship between dyad affiliation and coordination success (e.g. "Beta coefficient = 1.4, s.e.=0.4"). (Description)
- **TrainingTrials:** Number of training trials per individual/number of trials of experience with string-pull apparatus per individual prior to test phase. If unclear, then just code as "unclear". (Numerical)
- **SuccessCriterion1:** Description of criterion(s) for success for passing on to testing. (Description)
- **TestTrials:** Test trials per individual. If unclear, then just code as "unclear". (Numerical)
- **SuccessCriterion2:** Success criterion for success in coordination task. (Description)
- **RepeatedMeasures:** Repeated measures or not (Yes/No). (Binary categorical variable)
- **AnalysisLevel:** Level of analysis - if a paper does both, then add a new row for each dyad and code for each dyad (group/individual). (Binary categorical variable)
- **RopeLength:** Total length of rope used (metres). If total length is not given, then leave as "NA". (Numerical)

- **SampleSize:** Sample size/number of individual animals tested. (Numerical)
- **DelayTime:** Delay time used in delay task (s). If not specified then leave as "NA". (Numerical)
- **Av.SuccTrialsDelay:** Average number of successful trials in delay task. If not clear from paper, leave as "NA". (Numerical)
- **LengthOfTrials(m):** Length of test trials in minutes. (Numerical)
- **DelayOrSim:** Does study incorporate delay release, simultaneous release, neither or both simultaneous and delay release iterations of task? (simultaneous/delay/neither/both). (Categorical variable)
- **ApparatusDesc:** Description of apparatus. (Description)
- **ParticipatedBefore:** Have subjects participated in the task before? (Description)
- **AuthorConclusion:** Description of author conclusion from study. (Description)
- **ConclusionCode:** Did authors conclude species were successful or not (successful/not/NA). This variable is only applicable to studies that used a delay task. If authors did not use delay task, the code as "NA". (Categorical variable)
- **PublicationType:** Was publication a journal article, unpublished manuscript, PhD thesis etc. (Categorical variable)
- **PubStatus:** Whether or not the data comes from a study that is published in a journal or not. (Published/Unpublished) (Binary categorical variable)

Appendix G

Reply to Reviewer

Liam Keeble, Joel Wallenberg, Elizabeth E. Price

December 29, 2021

Once again we thank both the editor and reviewer for both their time and their helpful comments. We hope we have responded to them in full. Our responses are indicated by the bold text.

1 Reviewer 2

- The authors have completed the study in-line with the Stage 1 submission, with their noted change to the search after finding Animal Behaviour did not accept wild cards in the search string. This could have been overcome by using a search engine (e.g., Scopus, Web of Science) to perform the searches but the authors might not have had access.
- Overall I believe the authors have performed a useful study and generated a useful dataset. However I believe the inferences the authors make, and general structure of the results and discussion could be much improved to ensure the value of the study is communicated effectively and that the interpretation is justified.

1.1 Overall Structure

- I thought the overall results and discussion section could be made easier to follow, especially because it jumps between methodological challenges, quality control and analyses. Could subheadings be used corresponding to each of the objectives set out in the introduction to make the structure of the results clearer to the reader?
- **We have realigned the structure of both our results and discussion sections according to our aims for this study.**

1.2 Quality control

- The authors provided a well presented dataset of both the raw data and the quality control, but I was surprised not to see the reasons for the decisions in the quality control document not also included. While the

Stage 1 submission didn't explicitly state that the reasons and relevant text for the decisions to the quality control questions would be recorded, I thought this would have been performed. For example, the authors state that populations were rarely representative, but understanding how this decision was made is important for the quality of the dataset (i.e., have a column in the datafile with the same description from the full text extract and a comment on why the decision has been made). If these data weren't extracted, I think it would be really valuable for the authors to extract them and add them to the dataset for each question. Commenting on why the decision has been made for internal validity seems particularly important (and what criteria were screened for this), especially as the authors make the claim that "studies of animal coordination are largely internally valid". This is not necessarily true, rather what the authors did find was that one coder assessing a small number of string pulling tasks did not find reasons to doubt their internal validity.

- **Reasons for the decisions accompanied by extracts from the relevant papers have been included in the quality assessment data frame (qualityAssessment.csv). Where short extracts could not be determined as enough evidence for a reason, readers have been directed to the relevant sections of the paper being assessed. This is usually where we have concluded that a quality assessment criteria is absent from the text, and thus there is no evidence for that reason other than the absence of the criteria in the paper. For example, where studies have not identified confounds, we have noted that there is no evidence that researchers have done so and invite readers to assess the relevant sections for themselves. We have also clarified the limitations of the conclusions that we can draw based on a subjective method such as quality assessment (ll. 589-593)**

1.3 Conclusions

- Another conclusion the authors made surprised me. About publication bias, the authors said their analysis suggested little publication bias in studies assessing animal co-ordination. This surprised me as I know of unpublished studies in this area, and that the statistical power of their analyses to detect publication bias is very low. If their overall sample size is 17 included studies, what number of unpublished similar studies would mean that there is little publication bias? Even 4 or 5 unpublished studies would be a lot here.. If they wanted to make the conclusion about little publication bias then I think power analyses to see the degree of e.g. asymmetry their analyses could detect would be necessary. Alternatively I see no issue in highlighting how difficult it can be to assess publication in very heterogenous datasets with small sample sizes (this is the conclusion I would draw, with a strong prior that some studies will be missing).

- **We recognise that our inference came across too strong here. We did not mean to imply that we believed such an inference could be made on such a small scale analysis, but that one could only make such an inference if the analyses were valid (although we recognise that this was unclear in the text). We hope we have now clarified the limitations of our conclusions regarding our statistical tests of publication bias (ll. 621-633).**

1.4 Discussion

- The authors discussion focuses on co-ordination tasks and what their data can tell us about string-pulling. As the other reviewer at Stage 1 predicted, this might not be so much. I think a section of the discussion should be devoted to the challenges the authors faced in collecting and synthesising data in this project, as I feel this is one of the major contributions it makes.
- **A section has been added describing some difficulties we encountered whilst conducting the study (ll. 747-773).**

1.5 Minor Comments

- Reasons for exclusion that are given in the text (Lines 442 - 450) also be presented in the PRISMA diagram
- **Reasons have been included in the relevant box in figure 1.**
- Could the raw agreement also be included in Table 2, i.e., 5/5 statements extract the same, or 3/5? I think this would be easier for people to understand given the small number of double coded variables.
- **Raw agreement scores have been included in table 2.**
- Could the quality control statements also be written in Table 1
- **Quality control statements have been written in table 1.**